# Flow Density Control: Generative Optimization Beyond Entropy-Regularized Fine-Tuning

**Riccardo De Santi**
ETH Zurich
ETH AI Center
rdesanti@ethz.ch

**Marin Vlastelica**
ETH Zurich
ETH AI Center
marin.vlastelica@inf.ethz.ch

**Ya-Ping Hsieh**
ETH Zurich
yaping.hsieh@inf.ethz.ch

**Zebang Shen**
ETH Zurich
zebang.shen@inf.ethz.ch

**Niao He**
ETH Zurich
ETH AI Center
niaohe@ethz.ch

**Andreas Krause**
ETH Zurich
ETH AI Center
krausea@ethz.ch

## Abstract

Adapting large-scale foundational flow and diffusion generative models to optimize task-specific objectives while preserving prior information is crucial for real-world applications such as molecular design, protein docking, and creative image generation. Existing principled fine-tuning methods aim to maximize the expected reward of generated samples, while retaining knowledge from the pre-trained model via KL-divergence regularization. In this work, we tackle the significantly more general problem of optimizing general utilities beyond average rewards, including risk-averse and novelty-seeking reward maximization, diversity measures for exploration, and experiment design objectives among others. Likewise, we consider more general ways to preserve prior information beyond KL-divergence, such as optimal transport distances and Rényi divergences. To this end, we introduce **F**low **D**ensity **C**ontrol (FDC), a simple algorithm that reduces this complex problem to a specific sequence of simpler fine-tuning tasks, each solvable via scalable established methods. We derive convergence guarantees for the proposed scheme under realistic assumptions by leveraging recent understanding of mirror flows. Finally, we validate our method on illustrative settings, text-to-image, and molecular design tasks, showing that it can steer pre-trained generative models to optimize objectives and solve practically relevant tasks beyond the reach of current fine-tuning schemes.

## 1 Introduction

Large-scale generative modeling has recently seen remarkable advancements, with flow [30, 31] and diffusion models [51, 52, 23] standing out for their ability to produce high-fidelity samples across a wide range of applications, from chemistry [24] and biology [9] to robotics [8]. However, approximating the data distribution is insufficient for real-world applications such as scientific discovery [6, 59], where one typically wishes to generate samples optimizing specific utilities, e.g., molecular stability and diversity, while preserving certain information from a pre-trained model. This problem has recently been tackled via fine-tuning in the case where the utility corresponds

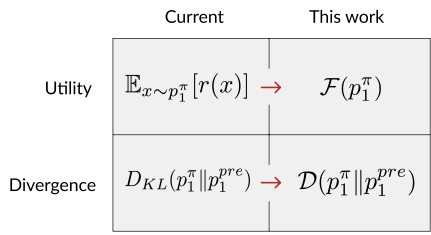

Figure 1: We extend the capabilities of current fine-tuning schemes from KL-regularized expected reward maximization (left) to the optimization of arbitrary distributional utilities $\mathcal{F}$ under general divergences $\mathcal{D}$ (right).

39th Conference on Neural Information Processing Systems (NeurIPS 2025).

to the expected reward of generated samples, and pre-trained model information is retained via KL-divergence regularization, as shown in Fig. 1 (left). Crucially, this specific fine-tuning problem can be solved via entropy-regularized control formulations [e.g., 14, 55, 53] with successful applications in real-world domains such as image generation [14], molecular design [56], or protein engineering [56].

Unfortunately, many practically relevant tasks cannot be captured by this formulation. For instance, consider the tasks of *risk-averse* and *novelty-seeking* reward maximization. In the former case, one wishes to steer the generative model toward distributions with controlled worst-case rewards, thereby improving validity and safety. In the latter case, one aims to control the upper tail of the reward distribution to maximize the probability of generating exceptionally promising designs, e.g., for scientific discovery. Other applications that cannot be captured via maximization of simple expectations include manifold exploration [12], model de-biasing [13], and optimal experimental design [38, 10] among others. Similarly, preserving prior information via a KL divergence has known drawbacks. For instance, it can lead to missing of low-probability yet valuable modes [29, 43], and it prevents from leveraging the geometry of the space even when this is known, e.g., in protein docking [9]. Replacing KL with alternative divergences can address these shortcomings. Driven by these motivations, in this work we aim to answer the following fundamental question (see Fig. 1):

> *How can we provably fine-tune a flow or diffusion model to optimize any user-specified utility while preserving prior information via an arbitrary divergence?*

Answering this would contribute to the algorithmic-theoretical foundations of *generative optimization*.

**Our approach** We tackle this challenge by first introducing the formal problem of *generative optimization via fine-tuning*. Then, we shed light on why this formulation is strictly more expressive than current fine-tuning problems [14, 53], and present a sample of novel practically relevant utilities and divergences (Sec. 3). Next, we introduce **F**low **D**ensity **C**ontrol (FDC), a simple sequential scheme that can fine-tune models to optimize general objectives beyond the reach of entropy-regularized control methods. This is achieved by leveraging recent machinery from Convex [20] and General Utilities RL [60] (Sec. 4). We provide rigorous convergence guarantees for the proposed algorithm in both a simplified scenario, via convex optimization analysis [42, 33], and in a realistic setting, by building on recent understanding of mirror flows [25] (Sec. 5). Finally, we provide an experimental evaluation of the proposed method, demonstrating its practical relevance on both synthetic and high-dimensional image and molecular generation tasks, showing how it can steer pre-trained models to solve tasks beyond the inherent limits of current fine-tuning schemes (Sec. 6).

**Our contributions** To sum up, in this work we contribute

- A formalization of the *generative optimization* problem, which extends current fine-tuning formulations beyond linear utilities and general divergences (Sec. 3).
- *Flow Density Control (FDC)*, a principled algorithm capable of optimizing functionals beyond the reach of current fine-tuning schemes based on entropy-regularized control/RL (Sec. 4).
- Convergence guarantees for the presented algorithm both under simplified and realistic assumptions leveraging recent understanding of mirror flows (Sec. 5).
- An experimental evaluation of FDC showcasing its practical relevance on both illustrative and high-dimensional text-to-image and molecular design tasks, showing how it can steer pre-trained models to solve tasks beyond the capabilities of current fine-tuning schemes. (Sec. 6).

## 2  Background and Notation

**General Notation.** We denote with $\mathcal{X} \subseteq \mathbb{R}^d$ an arbitrary set. Then, we indicate the set of Borel probability measures on $\mathcal{X}$ with $\mathbb{P}(\mathcal{X})$, and the set of functionals over the set of probability measures $\mathbb{P}(\mathcal{X})$ as $\mathbb{F}(\mathcal{X})$. Given an integer $N$, we define $[N] := \{1, \ldots, N\}$.

**Generative Flow Models.** Generative models aim to approximately sample novel data points from a data distribution $p_{data}$. Flow models tackle this problem by transforming samples $X_0 = x_0$ from a source distribution $p_0$ into samples $X_1 = x_1$ from the target distribution $p_{data}$[31, 17]. Formally, a *flow* is a time-dependent map $\psi : [0, 1] \times \mathbb{R}^d \to \mathbb{R}$ such that $\psi : (t, x) \to \psi_t(x)$. A *generative flow model* is a continuous-time Markov process $\{X_t\}_{0 \leq t \leq 1}$ obtained by applying a flow $\psi_t$ to $X_0 \sim p_0$ as $X_t = \psi_t(X_0)$, $t \in [0, 1]$, such that $X_1 = \psi_1(X_0) \sim p_{data}$. In particular, the flow $\psi$ can be defined by a *velocity field* $u : [0, 1] \times \mathbb{R}^d \to \mathbb{R}^d$, which is a vector field related to $\psi$ via the following ordinary differential equation (ODE), typically referred to as *flow ODE*:

$$\frac{\mathrm{d}}{\mathrm{d}t}\psi_t(x) = u_t(\psi_t(x)) \tag{1}$$

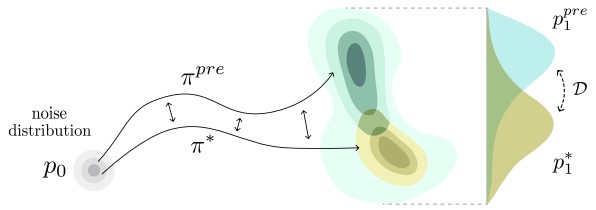
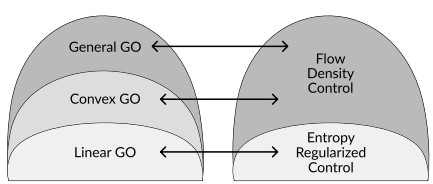

(a) Generative Optimization via Flow Model Fine-tuning.

(b) GO expressivity hierarchy

Figure 2: (2a) Pre-trained and fine-tuned policies inducing densities $p_1^{pre}$ and optimal density $p_1^*$ w.r.t. utility $\mathcal{F}$ and divergence $\mathcal{D}$. (2b) Expressivity and control hierarchy for generative optimization.

with initial condition $\psi_0(x) = 0$. A flow model $X_t = \psi_t(X_0)$ induces a probability path of *marginal densities* $p = \{p_t\}_{0 \leq t \leq 1}$ such that at time $t$ we have that $X_t \sim p_t$. Given a velocity field $u$ and marginal densities $p$, we say that $u$ generates the marginal densities $p = \{p_t\}_{0 \leq t \leq 1}$ if $X_t = \psi_t(X_0) \sim p_t$ for all $t \in [0, 1)$. This is the case if the pair $(u, p)$ satisfy the *Continuity Equation*:

$$\frac{\mathrm{d}}{\mathrm{d}t} p_t(x) + \mathrm{div}(p_t u_t)(x) = 0 \tag{2}$$

In this case, we denote by $p^u$ the probability path of marginal densities induced by the velocity field $u$. Flow matching [30, 32, 1, 31] can estimate a velocity field $u^\theta$ s.t. the induced marginal densities $p^{u_\theta}$ satisfy $p_0^{u_\theta} = p_0$ and $p_1^{u_\theta} = p_{data}$, where $p_0$ denotes the source distribution, and $p_{data}$ the target data distribution. Interestingly, diffusion models [52] (DMs) admit an equivalent ODE-based formulation with identical marginal densities to their original SDE dynamics [31, Chapter 10]. Consequently, although in this work we adopt the notation of flow models, our contributions carry over directly to DMs.

**Continuous-time Reinforcement Learning.** We formulate finite-horizon continuous-time reinforcement learning (RL) as a specific class of optimal control problems [57, 26, 54, 61]. Given a state space $\mathcal{X}$ and an action space $\mathcal{A}$, we consider the transition dynamics governed by the following ODE:

$$\frac{\mathrm{d}}{\mathrm{d}t} \psi_t(x) = a_t(\psi_t(x)) \tag{3}$$

where $a_t \in \mathcal{A}$ is a selected action. We consider a state space $\mathcal{X} := \mathbb{R}^d \times [0, 1]$, and denote by (Markovian) deterministic policy a function $\pi_t(X_t) := \pi(X_t, t) \in \mathcal{A}$ mapping a state $(x, t) \in \mathcal{X}$ to an action $a \in \mathcal{A}$ such that $a_t = \pi(X_t, t)$, and denote with $p_t^\pi$ the marginal density at time $t$ induced by policy $\pi$.

**Pre-trained Flow Models as an RL policy.** A pre-trained flow model with velocity field $u^{pre}$ can be interpreted as an action process $a_t^{pre} := u^{pre}(X_t, t)$, where $a_t^{pre}$ is determined by a continuous-time RL policy via $a_t^{pre} = \pi^{pre}(X_t, t)$ [12]. Therefore, we can express the flow ODE induced by a pre-trained flow model by replacing $a_t$ with $a^{pre}$ in Eq. (3), and denote the pre-trained model by its (implicit) policy $\pi^{pre}$, which induces a marginal density $p_1^{pre} := p_1^{\pi^{pre}}$ approximating $p_{data}$.

## 3 Formal Problem: a General Framework for Generative Optimization

In this section, we aim to formally introduce the general problem of generative optimization (GO) via fine-tuning. Formally, we wish to adapt a pre-trained generative flow model $\pi^{pre}$ to obtain a new model $\pi^*$ inducing an ODE:

$$\frac{\mathrm{d}}{\mathrm{d}t} \psi_t(x) = a_t^*(\psi_t(x)) \quad \text{with} \quad a_t^* = \pi^*(x, t), \tag{4}$$

such that instead of imitating the data distribution $p_{data}$, as typically in generative modeling, it induces a marginal density $p_1^{\pi^*}$ that maximizes a utility measure $\mathcal{F} : \mathbb{P}(\mathcal{X}) \to \mathbb{R}$, while preserving information from the pre-trained model $\pi^{pre}$ via regularization with an arbitrary divergence $\mathcal{D}(\cdot \| p^{pre})$. This algorithmic problem is illustrated in Fig. 2a, and formalized in the following.

### Generative Optimization via Flow Model Fine-Tuning

$$\arg\max_{\pi} \quad \mathcal{F}(p_1^\pi) - \alpha \mathcal{D}(p_1^\pi \| p_1^{pre}) \text{ s.t.} \frac{\mathrm{d}}{\mathrm{d}t} p_t(x) + \mathrm{div}(p_t a_t)(x) = 0 \text{ with } a_t = \pi(x, t) \tag{5}$$

| Application | Functional $\mathcal{F}$ / $\mathcal{D}$ | Linear GO | Non-Linear GO | |
| --- | --- | --- | --- | --- |
| | | | Convex | General |
| Reward optimization [14, 55] | $\mathbb{E}_{x\sim p^\pi}[r(x)]$ | ✓ | ✓ | ✓ |
| Manifold Exploration [12] Gen. model de-biasing | $\mathcal{H}(p^\pi) := - \underset{x\sim p^\pi}{\mathbb{E}}[\log p^\pi(x)]$ | ✗ | ✓ | ✓ |
| Risk-averse optimization | $\mathrm{CVaR}^r_\beta(p^\pi) := \underset{x\sim p^\pi}{\mathbb{E}}[r(x) \mid r(x) \leq \mathrm{q}^r_\beta(p^\pi)]$ | ✗ | ✓ | ✓ |
| | $\mathbb{E}_{x\sim p^\pi}[r(x)] - \mathbb{V}\mathrm{ar}(p^\pi)$ | ✗ | ✗ | ✓ |
| Novelty-seeking optimization | $\mathrm{SQ}^r_\beta(p^\pi) := \underset{x\sim p^\pi}{\mathbb{E}}[r(x) \mid r(x) \geq \mathrm{q}^r_\beta(p^\pi)]$ | ✗ | ✗ | ✓ |
| Optimal Experiment Design | $\mathrm{s}\left(\underset{x\sim p^\pi}{\mathbb{E}}[\Phi(x)\Phi(x)^\top - \lambda\mathbb{I}]\right)$ $\mathrm{s}(\cdot) \in \{\log\det(\cdot), -\mathrm{Tr}(\cdot)^{-1}, -\lambda_{max}(\cdot)\}$ | ✗ | ✓ | ✓ |
| Diverse modes discovery | $-\underset{z}{\mathbb{E}}[D_{KL}(p^{\pi,z} \| \underset{k}{\mathbb{E}} p^{\pi,k})]$ | ✗ | ✗ | ✓ |
| Log-Barrier Constrained Generation | $\mathbb{E}_{x\sim p^\pi}[r(x)] - \beta\log(\langle p^\pi, c\rangle - C)$ | ✗ | ✓ | ✓ |
| Kullback–Leibler divergence [14, 55] | $D_{KL}(p^\pi \| p^{pre}) = \int p^\pi(x)\log\frac{p^\pi(x)}{p^{pre}(x)}\,dx$ | ✓ | ✓ | ✓ |
| Rényi divergences | $D_\beta(p^\pi \| p^{pre}) := \frac{1}{\beta-1}\log\int (p^\pi(x))^\beta(p^{pre})^{1-\beta}\,dx$ | ✗ | ✗ | ✓ |
| Optimal Transport distances | $W_p(p^\pi \| p^{pre}) := \underset{\gamma\in\Gamma(p^\pi, p^{pre})}{\inf}\underset{(x,y)\sim\gamma}{\mathbb{E}}[d(x,y)^p]^{\frac{1}{p}}$ | ✗ | ✗ | ✓ |
| Maximum Mean Discrepancy | $\mathrm{MMD}_k(p^\pi \| p^{pre}) := \|\mu_{p^\pi} - \mu_{p^{pre}}\|, \mu_p := \underset{x\sim p}{\mathbb{E}}[k(x,\cdot)]$ | ✗ | ✓ | ✓ |

Table 1: Examples of practically relevant utilities $\mathcal{F}$ (blue) and divergences $\mathcal{D}$ (orange). Apx. A provides mathematical details and practical applications for each functional. Notice that besides $\mathcal{H}$, all non-linear functionals presented are novel in the context of fine-tuning of diffusion and flow models.

In this formulation, $\mathcal{F}$ and $\mathcal{D}$ are both functionals mapping the marginal density $p_1^\pi$ induced by policy $\pi$ to a scalar real number, namely $\mathcal{F}, \mathcal{D} : \mathbb{P}(\mathcal{X}) \to \mathbb{R}$. The constraint in Eq. (5) is the (*controlled*) Continuity Equation (see Eq. (2)), which relates the control policy $\pi$ to the induced marginal density $p_1^\pi$.

### 3.1 The sub-case of KL-regularized reward maximization via entropy-regularized control

Current fine-tuning schemes for flow generative models based on RL and control-theoretic formulations [e.g., 14, 55] aim to tackle the following problem, where we omit the flow constraint for clarity:

**Linear Generative Optimization via Flow Model Fine-Tuning**

$$\arg\max_\pi \quad \underset{x\sim p_1^\pi}{\mathbb{E}}[r(x)] - \alpha D_{KL}(p_1^\pi \| p_1^{pre}) \tag{6}$$

Crucially, the common problem in Eq. (6), which we denote by *Linear*[1] GO, is the specific sub-case of the generative optimization problem in Eq. (5), where the utility $\mathcal{F}$ is a linear functional corresponding to the expectation of a (reward) function $r : \mathcal{X} \to \mathbb{R}$, and $\mathcal{D}$ is the Kullback–Leibler divergence:

$$\mathcal{F}(p_1^\pi) = \langle p_1^\pi, r\rangle = \underset{x\sim p_1^\pi}{\mathbb{E}}[r(x)] \quad \text{and} \quad \mathcal{D}(p_1^\pi \| p_1^{pre}) = D_{KL}(p_1^\pi \| p_1^{pre}) \tag{7}$$

This specific fine-tuning problem can be solved via entropy-regularized (or relaxed) control [14].

### 3.2 Beyond Linear Generative Optimization: an Expressivity Viewpoint

Let $\mathcal{G}(p_1^\pi) = \mathcal{F}(p_1^\pi) - \alpha\mathcal{D}(p_1^\pi \| p_1^{pre})$ be the functional in Eq. (5). Then we denote by *Convex* GO the case where $\mathcal{G}$ is concave in $p_1^\pi$, and by *General* GO the case for arbitrary, possibly non-convex functionals [2]. In terms of expressivity **Linear GO** $\subset$ **Convex GO** $\subset$ **General GO**, as depicted in Fig. 2b (left). In Table 1 we classify into these tree tiers a sample of practically relevant utilities ($\mathcal{F}$, blue) and divergences ($\mathcal{D}$, orange). In Apx. A we report complete definitions and applications. Except for entropy [12] and KL, all non-linear functionals in Table 1 are to our knowledge explicitly used for the first time in the flow and diffusion model fine-tuning literature, while vastly employed in other areas. Moreover, the framework presented in this work for GO (Eq. 5) applies to any new choice of $\mathcal{F}$ or $\mathcal{D}$.

---

[1]For clarity, we adopt the term *linear* motivated by the linear utility even though the KL is non-linear.

[2]For clarity, we use the term *convex* GO, rather than concave GO, to denote the problem class where concave functionals are optimized.

---
**Algorithm 1** **F**low **D**ensity **C**ontrol (FDC)
---
1: **input:** $\mathcal{G}$ : general utility functional, $K$ : number of iterations, $\pi^{pre}$ : pre-trained flow generative model,
   $\{\eta_k\}_{k=1}^K$ regularization coefficients
2: **Init:** $\pi_0 := \pi^{pre}$
3: **for** $k = 1, 2, \ldots, K$ **do**
4:     Estimate: $\nabla_x g_k = \nabla_x \delta\mathcal{G}(p_1^{k-1})$
5:     Compute $\pi_k$ via first-order linear fine-tuning:

$$\pi_k \leftarrow \text{ENTROPYREGULARIZEDCONTROLSOLVER}(\nabla_x g_k, \eta_k, \pi_{k-1})$$

6: **end for**
7: **output:** policy $\pi := \pi_K$

---

Given the generality of generative optimization (Eq.(5)), a natural question arises: how can it be solved algorithmically? In the next section, we answer this by leveraging recent machinery from Convex [20] and General-Utilities RL [60], to derive a fine-tuning scheme that handles both convex and general GO, thus going beyond current entropy-regularized control methods, as illustrated in Fig. 2b (right).

## 4 Algorithm: Flow Density Control

In this section, we introduce **F**low **D**ensity **C**ontrol (FDC), see Alg. 1, which provably solves the generative optimization problem in Eq. (5) via sequential fine-tuning of the pre-trained model $\pi^{pre}$. To this end, we recall the notion of first variation of a functional over a space of probability measures [25]. A functional $\mathcal{G} \in \mathbb{F}(\mathcal{X})$, where $\mathcal{G} : \mathbb{P}(\mathcal{X}) \to \mathbb{R}$, has first variation at $\mu \in \mathbb{P}(\mathcal{X})$ if there exists a function $\delta\mathcal{G}(\mu) \in \mathbb{F}(\mathcal{X})$ such that for all $\mu' \in \mathbb{P}(\mathcal{X})$ it holds that:

$$\mathcal{G}(\mu + \epsilon\mu') = \mathcal{G}(\mu) + \epsilon\langle\mu', \delta\mathcal{G}(\mu)\rangle + o(\epsilon).$$

where the inner product has to be interpreted as an expectation. Intuitively, the first variation of $\mathcal{G}$ at $\mu$, namely $\delta\mathcal{G}(\mu)$, can be interpreted as an infinite-dimensional gradient in the space of probability measures. Given this notion, and a pair of generative models represented via policies $\pi$ and $\pi'$, we can now state the following *entropy-regularized first variation maximization* fine-tuning problem.

---

**Entropy-Regularized First Variation Maximization**

$$\arg\max_\pi \quad \langle\delta\mathcal{G}\left(p_1^{\pi'}\right), p_1^\pi\rangle - \eta D_{KL}(p_1^\pi \| p_1^{\pi'}) \tag{8}$$

---

Crucially, we can introduce a function $g : \mathcal{X} \to \mathbb{R}$ defined for all $x \in \mathcal{X}$ such that:

$$g(x) := \delta\mathcal{G}\left(p_1^{\pi'}\right)(x) \quad \text{and} \quad \mathbb{E}_{x \sim p^\pi}[g(x)] = \langle\delta\mathcal{G}\left(p_1^{\pi'}\right), p_1^\pi\rangle \tag{9}$$

As a consequence, by rewriting Eq. (8) expressing the first term via an expectation as shown in Eq. (9), it corresponds to a common Linear GO problem (see Eq. (6)), which can be optimized by utilizing established entropy-regularized control methods [e.g., 56, 14, 61].

We can finally present **F**low **D**ensity **C**ontrol (FDC), see Alg. 1, a mirror descent (MD) scheme [42] that reduces optimization of non-linear functionals $\mathcal{G}$ to a specific sequence of Linear GO problems. FDC takes three inputs: a pre-trained flow or diffusion model $\pi^{pre}$, the number of iterations $K$, and a sequence of regularization weights $\{\eta_k\}_{k=1}^K$. At each iteration, FDC first estimates the gradient of the functional first variation at the previous policy $\pi_{k-1}$, i.e., $\nabla_x \delta\mathcal{G}\left(p_1^{k-1}\right)$ (line 4). Then, it updates the flow model $\pi_k$ by solving the fine-tuning problem in Eq. (8) via an entropy-regularized control solver such as Adjoint Matching [14], using $\nabla_x g_k := \nabla_x \delta\mathcal{G}\left(p_1^{k-1}\right)$ as in Eq. (9) (line 5). Ultimately, it returns a final policy $\pi := \pi_K$. We report a detailed implementation of FDC in Apx. D.

**Gradient of first variation: computation and estimation.** Surprisingly, estimating $\nabla_x g_k$ in Alg. 1 (line 4) rarely requires density estimation. Among the functionals in Table 1, only the Rényi divergence does, for which one can leverage the recent Itô density estimator [50]. All other functionals admit straightforward plug-in or sample-based approximations detailed in Apx. A. As an illustrative example, in the following we showcase three examples from Table 1:

$$\nabla_x \delta\mathcal{Q}(p^\pi)(x) = \begin{cases} -\nabla_x \log p^\pi(x) & \text{Entropy } (\mathcal{H}) \\ \nabla_x r(x) \cdot \mathbf{1}\{r(x) \leq q_\beta^r(p^\pi)\} & \text{CVaR} \\ \nabla_x \phi^*(x) \text{ where } \phi^* = \arg\max_{\phi:\|\nabla_x\phi\|\leq 1}\langle\phi, p^\pi - p^{pre}\rangle & \text{Wasserstein-1 } (W_1) \end{cases}$$

Here $\mathcal{Q}$ denotes either a utility $\mathcal{F}$ or a divergence $\mathcal{D}$, and $q_\beta^r(p^\pi)$ is the $\beta$-quantile of $Z = r(X)$ with $X \sim p^\pi$ [47]. These gradients can be easily implemented. For entropy, the score term can be approximated via the score network in the case of diffusion models [12], and obtained via a known linear transformation of the learned velocity field in the case of flows [14, Eq.(8)]. For CVaR, any standard sample-based estimator of $q_\beta^r(p^\pi)$ [47] can be used. For Wasserstein-1, $\phi^*$ actually corresponds to the discriminator in Wasserstein-GAN, which can be learned with established methods [2]. In Apx. A, we report the gradient of the first variation for all functionals in Table 1, explain their practical estimation, and present a tutorial to derive the first variation of any new functionals not mentioned within Table 1.

Given the approximate gradient estimates and the generality of the objective functions, it is still unclear whether the proposed algorithm provably converges to the optimal flow model $\pi^*$. In the next section, we answer this question by developing a theoretical analysis via recent results on mirror flows [25].

# 5 Guarantees for Generative Optimization via Flow Density Control

In this section, we recast (5) as *constrained* optimization over stochastic processes, where the constraint is given by the Continuity Equation (2). This formulation enables the application of **mirror descent for constrained optimization** and the notion of *relative smoothness* [3]. In our framework, convergence speed is governed by: 1. the structural complexity of the functional $\mathcal{G}$ (cf. Section 4), 2. the accuracy of the estimator $g$ from (9), and 3. the quality of the oracle ENTROPYREGULARIZED-CONTROLSOLVER in Alg. 1. To handle these cases, we will analyze two representative regimes:

- **Idealized.** $\mathcal{G}$ is *concave*, and both $g$ and ENTROPYREGULARIZEDCONTROLSOLVER are exact. In this setting, classical results yield sharp step-size prescriptions and fast convergence rates.

- **General.** $\mathcal{G}$ is *non-concave*, with $g$ and the oracle subject to noise and bias. While fast convergence is generally out of reach [34, 27], convergence to a stationary point remains attainable under mild assumptions.

**Theoretical analysis: Idealized setting.** We now present a framework leading to convergence guarantees for FDC (i.e., Alg. 1) for *concave* functionals $\mathcal{G} \in \mathbb{F}(\mathcal{X})$. We start by recalling the notion of Bregman divergence induced by a functional $\mathcal{Q} \in \mathbb{F}(\mathcal{X})$ between densities $\mu, \nu \in \mathbb{P}(\mathcal{X})$, namely:
$$D_\mathcal{Q}(\mu \,\|\, \nu) := \mathcal{Q}(\mu) - \mathcal{Q}(\nu) - \langle \delta\mathcal{Q}(\nu), \mu - \nu \rangle$$

Next, we introduce two structural properties for our analysis.

**Definition 1** (Relative smoothness and relative strong concavity [33])**.** *Let* $\mathcal{G} : \mathbb{P}(\mathcal{X}) \to \mathbb{R}$ *a concave functional. We say that* $\mathcal{G}$ *is L-smooth relative to* $\mathcal{Q} \in \mathbb{F}(\mathcal{X})$ *over* $\mathbb{P}(\mathcal{X})$ *if* $\exists L$ *scalar s.t. for all* $\mu, \nu \in \mathbb{P}(\mathcal{X})$:
$$\mathcal{G}(\nu) \geq \mathcal{G}(\mu) + \langle \delta\mathcal{G}(\mu), \nu - \mu \rangle - L D_\mathcal{Q}(\nu \,\|\, \mu) \tag{10}$$
*and we say that* $\mathcal{G}$ *is l-strongly concave relative to* $\mathcal{Q} \in \mathbb{F}(\mathcal{X})$ *over* $\mathbb{P}(\mathcal{X})$ *if* $\exists l \geq 0$ *scalar s.t. for all* $\mu, \nu \in \mathbb{P}(\mathcal{X})$:
$$\mathcal{G}(\nu) \leq \mathcal{G}(\mu) + \langle \delta\mathcal{G}(\mu), \nu - \mu \rangle - l D_\mathcal{Q}(\nu \,\|\, \mu) \tag{11}$$

In the following, we interpret line (6) of FDC as a step of mirror ascent [42], and the KL divergence term as the Bregman divergence induced by an entropic mirror map $\mathcal{Q} = \mathcal{H}$, i.e., $D_{KL}(\mu, \nu) = D_\mathcal{H}(\mu \,\|\, \nu)$. We can finally state the following set of assumptions as well as the convergence guarantee for an arbitrary functional $\mathcal{G}(\cdot) = \mathcal{F}(\cdot) - \alpha\mathcal{D}(\cdot \,\|\, p^{pre}) \in \mathbb{F}(\mathcal{X})$.

**Assumption 5.1** (Exact estimation and optimization)**.** *We consider the following assumptions:*

1. *Exact estimation:* $\nabla_x \delta\mathcal{G}(p_1^k)$ *is estimated exactly* $\forall k \in [K]$.

2. *The optimization problem in Eq.* (8) *is solved exactly.*

**Theorem 5.1** (Convergence guarantee of Flow Density Control with concave functionals)**.** *Given Assumptions 5.1, fine-tuning a pre-trained model* $\pi^{pre}$ *via* FDC *(Algorithm 1) with* $\eta_k = L$ $\forall k \in [K]$*, leads to a policy* $\pi$ *inducing a marginal distribution* $p_1^\pi$ *such that:*

$$\mathcal{G}(p_1^*) - \mathcal{G}(p_1^\pi) \leq \frac{L - l}{K} D_{KL}(p_1^* \,\|\, p_1^{pre}) \tag{12}$$

*where* $p_1^* := p_1^{\pi^*}$ *is the marginal distribution induced by the optimal policy* $\pi^* \in \arg\max_\pi \mathcal{G}(p_1^\pi) := \mathcal{F}(p_1^\pi) - \alpha\mathcal{D}(p_1^\pi \,\|\, p_1^{pre})$.

Theorem 5.1 provides a fast convergence rate under a specific step-size choice ($\eta_k = L$). However, it critically depends on Assumption 5.1, which typically does not hold in practice. To address this limitation, we now consider a more general scenario where this key assumption is relaxed.

**Theoretical analysis: General setting.** Recall that $p_1^k := p_1^{\pi_k}$ represents the (stochastic) density produced by the ENTROPYREGULARIZEDCONTROLSOLVER oracle at the $k$-th step of FDC, and consider the following *mirror ascent* iterates, where $1/\lambda_k = \eta_k$ in Algorithm 1:

$$p_\sharp^k := \underset{p \in \mathbb{P}(\Omega_{pre})}{\arg\max} \quad \langle \delta\mathcal{G}\left(p_T^{\pi_{k-1}}\right), p \rangle - \frac{1}{\gamma_k} D_{KL}(p \,\|\, p_T^{\pi_{k-1}}) \tag{MD$_k$}$$

In realistic settings, where only noisy *and* biased approximations of (MD$_k$) are available, it is essential to quantify the deviations from the idealized iterates in (MD$_k$). To this end, denote by $\mathcal{T}_k$ the filtration up to step $k$, and consider the decomposition of the oracle into its *noise* and *bias* parts:

$$b_k := \mathbb{E}\left[\delta\mathcal{G}(p_T^{\pi_k}) - \delta\mathcal{G}(p_\sharp^k) \,|\, \mathcal{T}_k\right], \qquad U_k := \delta\mathcal{G}(p_T^{\pi_k}) - \delta\mathcal{G}(p_\sharp^k) - b_k \tag{13}$$

Conditioned on $\mathcal{T}_k$, $U_k$ has zero mean, while $b_k$ captures the *systematic* error. We then impose:

**Assumption 5.2** (Noise and Bias). *The following events happen almost surely:*

$$\|b_k\|_\infty \to 0, \qquad \sum_k \mathbb{E}\left[\gamma_k^2\left(\|b_k\|_\infty^2 + \|U_k\|_\infty^2\right)\right] < \infty, \qquad \sum_k \gamma_k \|b_k\|_\infty < \infty \tag{14}$$

The first condition is a *necessary* requirement for convergence since when violated, it is easy to construct scenarios where no practical algorithm can solve the generative optimization problem. The second and third inequalities manage the trade-off between *accuracy* of the approximate oracle ENTROPYREGULARIZEDCONTROLSOLVER and *aggressiveness* of the step sizes, $\gamma_k$. Intuitively, lower noise and bias in the oracle enable the use of larger step sizes. To this end, Assumption 5.2 provides a concrete criterion that guarantees the success of finding the optimal policy with probability one.

> **Theorem 5.2** (Convergence guarantee of Flow Density Control for general functionals). *Given the Robbins-Monro step-size rule: $\sum_k \gamma_k = \infty, \sum_k \gamma_k^2 < \infty$, under Assumption 5.2 and technical assumptions (see Appendix C), the sequence of marginal densities $p_1^k$ induced by the iterates $\pi_k$ of Algorithm 1 converges weakly to a stationary point $\tilde{p}_1$ of $\mathcal{G}$ almost surely, formally: $p_1^k \rightharpoonup \tilde{p}_1$ a.s..*

## 6 Experimental Evaluation

We analyze the ability of **F**low **D**ensity **C**ontrol (FDC) to induce policies optimizing complex non-linear objectives, and compare its performance with Adjoint Matching (AM) [14], a classic fine-tuning method. We present two types of experiments: (i) Illustrative settings to provide insights via visual interpretability, and (ii) High-dimensional real-world applications, namely (a) novelty-seeking molecular design for single-point energy minimization [18], and (b) manifold exploration for text-to-image *creative bridge design* generation. Additional details are provided in Apx. E.

**Risk-averse reward maximization for better worst-case validity or safety.** We fine-tune a pre-trained policy $\pi^{pre}$ (see Fig. 3a) by optimizing the CVaR$_\beta$ utility i.e., expected outcome in the $\beta$-worst-case (see Tab. 1) with KL regularization, and costs interpreted as negative rewards. The cost has three regions: a high-cost plateau (dark orange), where the initial density lies; a moderate-cost left area (light orange); and a predominantly low-cost right zone (yellow) punctuated by narrow, but catastrophic red-stripes. As shown in Fig. 3b, AM moves the model density into the yellow region, lowering average cost but exposing it to rare extreme costs. In contrast, FDC, run with $K = 2$ iterations and $\beta = 0.01$, successfully steers density into the safer, moderate-cost area, cutting the 1%-worst-case cost from 288.2 achieved by AM to 90.0, well below the initial 262.5, as shown in Fig. 3c and 3d.

**Novelty-seeking reward maximization for discovery.** We fine-tune a pre-trained policy $\pi^{pre}$ to maximize the SQ$_\beta$ utility, i.e., expected outcome in the $\beta$-best-case (see Tab. 1). The reward shown in Fig. 3e has a moderately high-reward left area (light gray), a medium-reward central plateau (darker gray) where the initial density lies, and a low-reward right region (black) with sparse, extreme-reward spikes depicted by thin white lines. As shown in Fig. 3f, AM drifts the density into the safer left basin — improving the average reward but only reaching a best-1% expected reward of 55.5, as shown in Fig. 3g and Fig. 3h. In contrast, FDC, run for $K = 2$ iterations and $\beta = 0.99$, pushes the density rightwards, elevating the top-1% reward to 596.1 (see Fig. 3h) — far above both AM and the initial 66.6.

**Reward maximization regularized via optimal transport distance.** We fine-tune the pre-trained model with density in Fig. 3i to maximize a reward function that increases moving top right. We consider two $W_1$ distances induced by two ground metrics: $d_A$, which makes vertical moves more costly than horizontal ones, and $d_B$, which does the opposite. Under $d_A$, both AM and the OT-regularized model reach an expected reward of 35.0, but FDC-A incurs only $W_1^A = 1.95$ versus 4.67 for AM,

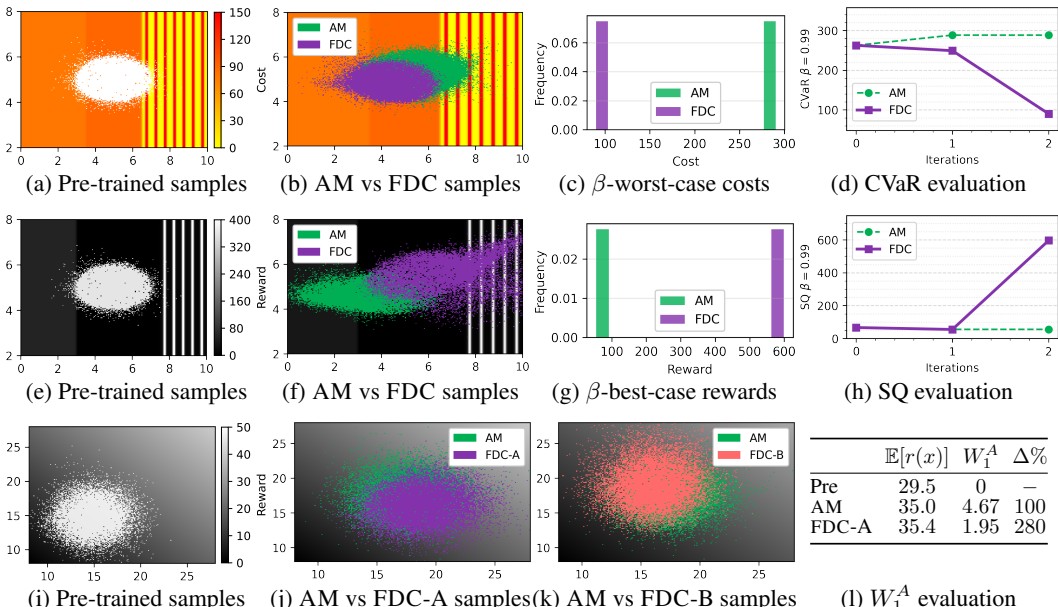

Figure 3: Illustrative settings with visually interpretable results. (top) Risk-averse reward maximization for valid or safe generation, (mid) Novelty-seeking reward maximization for discovery, (bottom) Expected rewards maximization under optimal transport distance regularization. Crucially, FDC can optimize well these complex objectives, while AM [14], a classic fine-tuning scheme, fails at this.

and achieves a mean shift that is $280\%$ larger in the horizontal than in the vertical direction (Fig. 3j and Tab. 3l). By contrast, FDC-B under $d_B$ preferentially shifts the density upward (Fig. 3k).

**Conservative manifold exploration.** We tackle manifold exploration [12] by fine-tuning a pre-trained model $\pi^{pre}$ to maximize the entropy utility ($\mathcal{H}$ in Tab. 1) under a KL regularization of strength $\alpha$, a capability not possible with prior methods [12]. As in previous work, we consider the common setting where the pre-trained model density $p_1^{pre}$ concentrates most of its mass in a specific region as shown in Fig. 4a, where $N = 10000$ samples are shown. By fine-tuning $\pi^{pre}$ via FDC, the density of the fine-tuned model shifts into low-coverage areas (see Fig. 4b and 4c). In particular, Fig. 4d demonstrates that reducing $\alpha$ from 0.5 to 0.0 yields progressively higher Monte Carlo entropy estimates (7.00 at $\alpha = 0.5$, 7.14 at $\alpha = 0$), thus enabling control of the trade-off between preserving the original distribution and exploring novel regions, a capability not supported by prior methods [12].

**Molecular design for single-point energy minimization.** We fine-tune FlowMol [15], pre-trained on QM9 [46], to discover molecules minimizing the single-point total energy computed via extended tight-binding at the GFN1-xTB level of theory [18]. Concretely, we maximize the negative energy. We do not aim to maximize the average sample reward, but rather that of the top $0.2\%$ samples. We employ FDC with novelty-seeking SQ utility (see Tab. 1) with $\beta = 0.998$, and make 2 gradient steps per $K = 10$ iterations. We compare it with AM run for 240 steps. Fig. 4j shows that while AM generates better samples in average (namely 29.1 over 27.5 of FDC), the average quality of the top $0.2\%$ molecules, indicated by $SQ_\beta$ is higher for FDC than for AM (namely 41.8 over 39.7 of AM). This confirms (see Fig. 4i and 4h) that FDC can sacrifice the average reward to generate a few truly high-reward designs.

**Text-to-image bridge designs conservative exploration.** We perform manifold exploration by fine-tuning Stable Diffusion (SD) 1.4 [49] with prompt "A creative bridge design.". To this end, we maximize the KL-regularized entropy (see Tab. 1) with $\alpha = 0.001$ via FDC for $K = 2$ steps. As a diversity metric, we utilize the Vendi score [19] with cosine similarity kernel on the extracted CLIP [21] features from a sample of 100 images and compared it to the baseline pre-trained model in Fig. 4g. Beyond increasing the Vendi score, FDC also increases the CLIP score of the initial model.

# 7 Related Works

**Flow and diffusion models fine-tuning via optimal control.** Recent works have framed fine-tuning of diffusion and flow models to maximize expected reward under KL regularization as an entropy-regularized optimal control problem [e.g., 55, 53, 56, 14]. Crucially, as shown in Sec. 3, the problem tackled by these studies is the specific sub-case of generative optimization (Eq. (5)),

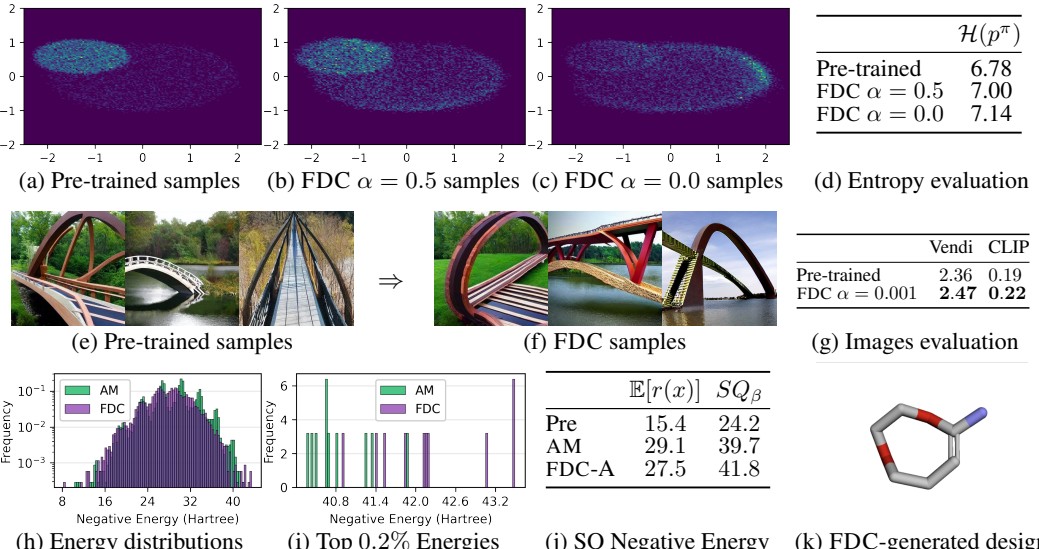

|  | $\mathcal{H}(p^\pi)$ |
|---|---|
| Pre-trained | 6.78 |
| FDC $\alpha = 0.5$ | 7.00 |
| FDC $\alpha = 0.0$ | 7.14 |

(a) Pre-trained samples    (b) FDC $\alpha = 0.5$ samples    (c) FDC $\alpha = 0.0$ samples    (d) Entropy evaluation

|  | Vendi | CLIP |
|---|---|---|
| Pre-trained | 2.36 | 0.19 |
| FDC $\alpha = 0.001$ | **2.47** | **0.22** |

(e) Pre-trained samples        (f) FDC samples        (g) Images evaluation

|  | $\mathbb{E}[r(x)]$ | $SQ_\beta$ |
|---|---|---|
| Pre | 15.4 | 24.2 |
| AM | 29.1 | 39.7 |
| FDC-A | 27.5 | 41.8 |

(h) Energy distributions    (i) Top 0.2% Energies    (j) SQ Negative Energy    (k) FDC-generated design

Figure 4: (top) Illustrative manifold exploration experiment via KL-regularized entropy maximization, (mid) High-dimensional manifold exploration via text-to-image model fine-tuning for prompt "A creative bridge design". Left: images from pre-trained model, Right: images from model fine-tuned via FDC, with higher diversity as indicated by a higher Vendi score. (bottom) Novelty-seeking molecular design for Energy (kcal/mol) maximization by fine-tuning FlowMol [15]. FDC shows enhanced control capabilities for optimizing such complex objectives than AM, a classic fine-tuning scheme.

where the utility $\mathcal{F}$ is linear, and $\mathcal{D} = D_{KL}$. In this work, we propose a principled method with guarantees for the far more general class of non-linear utilities and divergences beyond KL, including the ones listed in Tab. 1. The framework introduced has strictly higher expressive power and control capabilities for fine-tuning generative model (see Sec. 3). This renders possible to tackle relevant tasks e.g., scientific discovery, beyond the capabilities of the aforementioned fine-tuning schemes.

**Convex and General Utilities Reinforcement Learning.** Convex and General (Utilities) RL [20, 58, 60] generalizes RL to the case where one wishes to maximize a concave [20, 58], or general [60, 4] functional of the state distribution induced by a policy over a dynamical system's state space. The introduced generative optimization problem (in Eq. (5)) is related, with $p_1^\pi$ representing the state distribution induced by policy $\pi$ over a subset of the state space. Recent works tackled the finite samples budget setting [e.g., 41, 39, 40, 44, 11]. Ultimately, to our knowledge, this is the first work leveraging an algorithmic scheme resembling General RL for the practically relevant task of generative optimization of general non-linear functionals via fine-tuning of diffusion and flow models.

**Optimization over probability measures via mirror flows.** Recently, there has been a growing interest in building theoretical guarantees for optimization problems over spaces of probability measures in a variety of applications. These include GANs [25], optimal transport [3, 28, 27], kernelized methods [16], and manifold exploration [12]. We present the first use of this framework to establish guarantees for the generative optimization problem in Eq. (5). This novel link to probability-space optimization sheds new light on large-scale flow and diffusion models fine-tuning.

# 8 Conclusion

This work tackles the fundamental challenge of fine-tuning pre-trained flow and diffusion generative models on arbitrary task-specific utilities and divergences while retaining prior knowledge. We introduce a unified generative optimization framework that strictly generalizes existing formulations and propose a rich class of new practically relevant objectives. We then propose Flow Density Control, a mirror-descent algorithm that reduces complex generative optimization to a sequence of standard fine-tuning steps, each solvable by scalable off-the-shelf methods. Leveraging convex analysis and recent advances in mirror flows theory, we prove convergence under general conditions. Empirical results on synthetic benchmarks, molecular design, and image generation, demonstrate that our approach can steer pre-trained models to optimize objectives beyond the reach of current fine-tuning techniques. As for limitations, while our framework is general, future work will need to assess to what extent the flexibility in selecting utilities and divergences yields concrete gains in specific applications.

## Acknowledgements

This publication was made possible by the ETH AI Center doctoral fellowship to Riccardo De Santi, and postdoctoral fellowship to Marin Vlastelica. The project has received funding from the Swiss National Science Foundation under NCCR Catalysis grant number 180544 and NCCR Automation grant agreement 51NF40 180545.

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

# Contents

# A Functionals and Derivation of Gradients of First-order Variations

## A.1 Overview of utilities and divergences in Table 1

In the following, we report the missing details for the functionals presented within Table 1, and discuss some possible applications.

**Manifold Exploration and Generative Model De-biasing** As mentioned within Sec. 3, maximization of the entropy functional as been recently introduced as a principled objective for manifold exploration [12]. Moreover, we wish to point out that it can be interpreted also from the viewpoint of de-biasing a prior generative model to re-distribute more uniformly its density while preserving a certain notion of support, e.g., via sufficient KL-divergence regularization.

**Risk-averse and Novelty-seeking reward maximization** A definition of $q_\beta^r$ can be found below, explanations of these utilities can be found in Sec. 1, and experimental illustrative examples are provided in Sec. 6.

**Optimal Experiment Design** The task of Optimal Experimental Design (OED) [7] involves choosing a sequence of experiments so as to minimize some uncertainty metric for an unknown *quantity of interest* $f : \mathcal{X} \to \mathbb{R}$, where $\mathcal{X}$ is the set of all possible experiments. From a probabilistic standpoint, an optimal design may be viewed as a probability distribution over $\mathcal{X}$, prescribing how frequently each experiment should be performed to achieve maximal reduction in uncertainty about $f$ [45]. This problem has been recently studied in the case where $f$ is an element of a reproducing kernel Hilbert space (RKHS), i.e., $f \in \mathcal{H}_k$, induced by a known kernel $k(x, x') = \Phi(x)^\top \Phi(x')$ where $x, x' \in \mathcal{X}$ [37]. Given this setting, one might aim to acquire information about $f$ according to different *criteria* captured by the scalarization function $s(\cdot)$ [38]. In particular, in Table 1, we report three illustrative choices for $s$:

- D-design: $\log \det(\cdot)$ (Information)
- A-design: $-\mathrm{Tr}(\cdot)$ (Parameter error)
- E-design: $\lambda_{max}(\cdot)$ (Worst projection error)

as reported in previous work [Table 1 38].

**Diverse Mode Discovery** This objective corresponds to a re-interpretation of the Diverse Skill Discovery objective introduced in the context of Reinforcement Learning [58]. Consider the case where it is given a discrete and finite set $\mathcal{S}$ of symbols interpretable as latent variables, which can be leveraged to (exactly or approximately) perform conditional generation. This objective captures the task of assuring maximal diversity, in terms of KL divergence between the different conditional components, represented as $p^{\pi,k}$ with $k \in \mathcal{S}$.

**Log-barrier constrained generation** This formulation can be found within the General Utilities RL literature [60]. In particular, here we show the case where constraints are enforced via a log-barrier function, namely $\log(\cdot)$. Nonetheless, the functional presented in Table 1 remains meaningful for general penalty functions.

**Optimal transport distances** OT distances within Table 1 and their relative notation are introduced below in the context of their first variation computation.

**Maximum Mean Discrepancy** Here $k$ denotes a positive-definite kernel, which measures similarity between two points in sample space. Moreover, $\mu_p$ denotes a kernel mean embedding of distribution $p$ [36]. In terms of applications, choosing a proper kernel $k$ could render possible to preserve specific structure of the initial pre-trained model that would be otherwise lost via KL regularization.

## A.2 A brief tutorial on first variation derivation

In this work, we focus on the functionals that are Fréchet differentiable: Let $V$ be a normed spaces. Consider a functional $F : V \to \mathbb{R}$. There exists a linear operator $A : V \to \mathbb{R}$ such that the following

limit holds

$$\lim_{\|h\|_V \to 0} \frac{|F(f+h) - F(f) - A[h]|}{\|h\|_V} = 0. \tag{15}$$

We further assume that $V$ admits certain structure such that every element in its dual space (the space of bounded linear operator on $V$) admits some compact representation. For example, when $V$ is the set of compact-supported continuous bounded functions, there exists a unique positive Borel measure $\mu$ with the same support, which can be identified as the linear functional. We denote this element as $\delta F[f]$ such that $\langle \delta F[f], h \rangle = A[h]$. Sometimes we also denote it as $\frac{\delta F}{\delta f}$. We will refer to $\delta F[f]$ as the first-order variation of $F$ at $f$.

In this section, we briefly review strategies for deriving the first-order variation of two broad classes of functionals: those defined in closed form with respect to the density (e.g., expectation and entropy) and those defined via variational formulations (e.g., CVaR, Wasserstein distance, and MMD).

- **Category 1: Functional defined in a closed form w.r.t. the density.** For this class of functionals, the first-order variations can typically be computed using its definition and chain rule.

  With definition (15) in mind, we can try to calculate the first-order variation of the mean functional. Consider a continuous and bounded function $r : \mathbb{R}^d \to \mathbb{R}$ and a probability measure $\mu$ on $\mathbb{R}^d$. Consider the functional $F(\mu) = \int r(x)\mu(x)dx$. We have

  $$|F(\mu + \delta\mu) - F(\mu) - \langle r, \delta\mu \rangle| = 0. \tag{16}$$

  We therefore obtain $\delta F[\mu] = r$ for all $\mu$. We will compute the first-order variations for other functionals in the next subsection.

- **Category 2: Functionals defined through a variational formulation.** Another important subclass of functionals considered in this paper is the ones defined via a variational problem

  $$F[f] = \sup_{g \in \Omega} G[f, g], \tag{17}$$

  where $\Omega$ is a set of functions or vectors independent of the choice of $f$, and $g$ is optimized over the set $\Omega$. We will assume that the maximizer $g^*(f)$ that reaches the optimal value for $G[f, \cdot]$ is unique (which is the case for the functionals considered in this project). It is known that one can use the Danskin's theorem (also known as the envelope theorem) to compute

  $$\frac{\delta F[f]}{\delta f} = \partial_f G[f, g^*(f)], \tag{18}$$

  under the assumption that $F$ is differentiable [35].

### A.3 Derivation of gradients of first-order variation for functionals in Table 1

- **Risk-Averse Optimization (Category 2)** Recall that $q_\beta^r(p^\pi) = \sup\{v \in \mathbb{R} | F_Z(v) \le \beta\}$, where the random variable $Z$ is defined as $Z = r(x)$ with $x \sim p^\pi(x)$. From [48], we have

  $$\mathrm{CVaR}_\beta^r(p^\pi) = \mathbb{E}[r(x)|r(x) \le q_\beta^r(p^\pi)] = \beta \inf_\zeta \left\{ \zeta + \frac{1}{\beta}\mathbb{E}\left[\min\{r(x) - \zeta, 0\}\right] \right\}.$$

  Moreover, we have $\zeta^*$ that solves the above optimization problem is exactly $\zeta^* = q_\beta^r(p^\pi)$. By Danskin's theorem, one has (in a weak sense)

  $$\frac{\delta\mathrm{CVaR}_\beta^r(p^\pi)}{\delta p^\pi} = \beta \min\{r(x) - q_\beta^r(p^\pi), 0\}. \tag{19}$$

- **Risk-Seeking Optimization (Category 2)** Recall that $q_\beta^r(p^\pi) = \sup\{v \in \mathbb{R} | F_Z(v) \le \beta\}$, where the random variable $Z$ is defined as $Z = r(x)$ with $x \sim p^\pi(x)$. From [48], we have

  $$\mathrm{SQ}_\beta^r(p^\pi) = \mathbb{E}[r(x)|r(x) \ge q_\beta^r(p^\pi)] = (1-\beta) \inf_\zeta \left\{ \zeta + \frac{1}{1-\beta}\mathbb{E}\left[\max\{r(x) - \zeta, 0\}\right] \right\}.$$

  Moreover, we have $\zeta^*$ that solves the above optimization problem is exactly $\zeta^* = q_\beta^r(p^\pi)$. By Danskin's theorem, one has (in a weak sense)

  $$\frac{\delta\mathrm{SQ}_\beta^r(p^\pi)}{\delta p^\pi} = (1-\beta) \max\{r(x) - q_\beta^r(p^\pi), 0\}. \tag{20}$$

| APPLICATION | FUNCTIONAL $\mathcal{F}$ / $\mathcal{D}$ | FIRST-ORDER VARIATION | DENSITY CONTROL | |
| --- | --- | --- | --- | --- |
| | | | CONVEX | GENERAL |
| REWARD OPTIMIZATION [14, 55] | $\mathbb{E}_{x\sim p^\pi}[r(x)]$ | $r$ | ✓ | ✓ |
| MANIFOLD EXPLORATION GEN. MODEL DE-BIASING | $\mathcal{H}(p^\pi) := -\mathbb{E}_{x\sim p^\pi}[\log p^\pi(x)]$ | $-1 - \log p^\pi$ | ✓ | ✓ |
| RISK-AVERSE OPTIMIZATION | $\mathrm{CVaR}_\beta^r(p^\pi) := \mathbb{E}_{x\sim p^\pi}[r(x) \mid r(x) \leq \mathrm{q}_\beta^r(p^\pi)]$ | $\beta \min\{r(x) - q_\beta^r(p^\pi), 0\}$ | ✓ | ✓ |
| | $\mathbb{E}_{x\sim p^\pi}[r(x)] - \mathbb{V}\mathrm{ar}(p^\pi)$ | $r(x) - (r(x)^2 - 2\mathbb{E}_{x\sim p^\pi}[r(x)]r(x))$ | ✗ | ✓ |
| RISK-SEEKING OPTIMIZATION | $\mathrm{SQ}_\beta^r(p^\pi) := \mathbb{E}_{x\sim p^\pi}[r(x) \mid r(x) \geq q_\beta^r(p^\pi)]$ | $(1-\beta)\max\{r(x) - q_\beta^r(p^\pi), 0\}$ | ✗ | ✓ |
| OPTIMAL EXPERIMENT DESIGN | $\mathrm{s}(\mathbb{E}_{x\sim p^\pi}[\Phi(x)\Phi(x)^\top - \lambda\mathbb{I}])$ $\mathrm{s}(\cdot) \in \{\log\det(\cdot), -\mathrm{Tr}(\cdot)^{-1}, -\lambda_{max}(\cdot)\}$ | SEE EQUATION (30) | ✓ | ✓ |
| DIVERSE MODES DISCOVERY | $-\mathbb{E}_z[D_{KL}(p^{\pi,z} \| \mathbb{E}_k p^{\pi,k})]$ | SEE EQUATION (32) | ✗ | ✓ |
| LOG-BARRIER CONSTRAINED GENERATION | $\mathbb{E}_{x\sim p^\pi}[r(x)] - \beta\log(\langle p^\pi, c\rangle - C)$ | SEE EQUATION (31) | ✓ | ✓ |
| KULLBACK–LEIBLER DIVERGENCE | $D_{KL}(p^\pi \| p^{pre}) = \int p^\pi(x)\log\frac{p^\pi(x)}{p^{pre}(x)}dx$ | $1 + \log p^\pi - \log p^{pre}$ | ✓ | ✓ |
| RÉNYI DIVERGENCES | $D_\beta(p^\pi \| p^{pre}) := \frac{1}{\beta-1}\log\int(p^\pi(x))^\beta(p^{pre}(x))^{1-\beta}dx$ | $\frac{\beta}{\beta-1}\left(\int\left(\frac{p}{q}\right)^\beta dq(x)\right)^{-1}\left(\frac{p}{q}\right)^{\beta-1}$ | ✓ | ✓ |
| OPTIMAL TRANSPORT DISTANCES | $W_p(p^\pi \| p^{pre}) := \inf_{\gamma\in\Gamma(p^\pi,p^{pre})}\mathbb{E}_{(x,y)\sim\gamma}[d(x,y)^p]^{\frac{1}{p}}$ | SEE EQUATION (29) | ✓ | ✓ |
| MAXIMUM MEAN DISCREPANCY | $\mathrm{MMD}_k(p^\pi, p^{pre}) := \|\mu_{p^\pi} - \mu_{p^{pre}}\|, \mu_p := \mathbb{E}_{x\sim p}[k(x,\cdot)]$ | $\arg\max_{\phi\in\mathcal{H}}\langle\phi, p^\pi - p^{pre}\rangle$ | ✓ | ✓ |

Table 2: Examples of practically relevant utilities $\mathcal{F}$ (blue) and divergences $\mathcal{D}$ (orange), and their first-order variations.

- **Rényi Divergence (Category 1)** Recall the definition of Rényi Divergence

$$D_\beta(p\|q) = \frac{1}{\beta-1}\log\int\left(\frac{p}{q}\right)^\beta dq(x). \tag{21}$$

We ignore higher-order terms like $O((\delta p)^2)$.

$$D_\beta(p+\delta p\|q) - D_\beta(p\|q) = \frac{1}{\beta-1}\log\frac{\int\left(\frac{p+\delta p}{q}\right)^\beta dq(x)}{\int\left(\frac{p}{q}\right)^\beta dq(x)} \tag{22}$$

$$= \frac{1}{\beta-1}\log\frac{\int\left(\frac{p}{q}\right)^\beta + \beta\left(\frac{p}{q}\right)^{\beta-1}\frac{\delta p}{q}dq(x)}{\int\left(\frac{p}{q}\right)^\beta dq(x)} \tag{23}$$

$$= \frac{1}{\beta-1}\log 1 + \frac{\int\beta\left(\frac{p}{q}\right)^{\beta-1}\frac{\delta p}{q}dq(x)}{\int\left(\frac{p}{q}\right)^\beta dq(x)} \tag{24}$$

$$= \frac{1}{\beta-1}\frac{\int\beta\left(\frac{p}{q}\right)^{\beta-1}\frac{\delta p}{q}dq(x)}{\int\left(\frac{p}{q}\right)^\beta dq(x)} \tag{25}$$

$$\frac{\delta}{\delta p}R_\beta(p,q) = \frac{\beta}{\beta-1}\left(\int\left(\frac{p}{q}\right)^\beta dq(x)\right)^{-1}\left(\frac{p}{q}\right)^{\beta-1} \tag{26}$$

- **Optimal transport and Wasserstein-p distance (Category 2)** Consider the optimal transport problem

$$\mathrm{OT}_c(u,v) = \inf_\gamma\left\{\int\int c(x,y)d\gamma(x,y) : \int\gamma(x,y)dx = u(y), \int\gamma(x,y)dy = v(x)\right\} \tag{27}$$

where

$$\Gamma = \left\{ \gamma : \int \gamma(x,y)dx = u(y), \int \gamma(x,y)dy = v(x) \right\}$$

It admits the following equivalent dual formulation

$$\mathrm{OT}_c(u,v) = \sup_{f,g} \left\{ \int f du + \int g dv : f(x) + g(y) \le c(x,y) \right\} \tag{28}$$

By taking $c(x,y) = \|x - y\|^p$, we recover $\mathrm{OT}_c(u,v) = W_p(u,v)^p$. Let $f^*$ and $g^*$ be the solution to the above dual optimization problem. From the Danskin's theorem, we have

$$\frac{\delta}{\delta u} W_p(u,v)^p = f^*. \tag{29}$$

In the special case of $p = 1$, we know that $g^* = -f^*$ (note that the constraint can be equivalently written as $\|\nabla f\| \le 1$), in which case $f^*$ is typically known as the critic in the WGAN framework.

- **Optimal Experiment Design. (Category 1)** We take $s(M) = \log \det(M)$ as example. By chain rule, we have

$$\delta F[p^\pi] = \mathrm{Tr}\left[ \left( \underset{x \sim p^\pi}{\mathbb{E}} [\Phi(x)\Phi(x)^\top - \lambda \mathbb{I}] \right)^{-1} \left( \Phi(x)\Phi(x)^\top - \lambda \mathbb{I} \right) \right]. \tag{30}$$

- **Log-Barrier Constrained Generation. (Category 1)** By chain rule, we obtain

$$\delta F[p^\pi] = r - \frac{\beta c}{\langle p^\pi, c \rangle - C}. \tag{31}$$

- **Diverse modes discovery. (Category 1)** By chain rule, we obtain

$$\begin{aligned}
\frac{\delta F}{\delta p^{\pi,z}} &= -\frac{\delta}{\delta p^{\pi,z}} \mathbb{E}_z \left[ \int p^{\pi,z} \log p^{\pi,z} dx - \int p^{\pi,z} \log \left( \mathbb{E}_k[p^{\pi,k}] \right) dx \right] \\
&= -\mathbb{E}_z \left[ \frac{\delta}{\delta p^{\pi,z}} \left( \int p^{\pi,z} \log p^{\pi,z} dx \right) - \frac{\delta}{\delta p^{\pi,z}} \left( \int p^{\pi,z} \log \left( \mathbb{E}_k[p^{\pi,k}] \right) dx \right) \right] \\
&= -\mathbb{E}_z \left[ \log p^{\pi,z} + 1 - \log \left( \mathbb{E}_k[p^{\pi,k}] \right) - \frac{p^{\pi,z}}{\mathbb{E}_k[p^{\pi,k}]} \right] \tag{32}
\end{aligned}$$

- **Entropy. (Category 1)** As a first example, consider the entropy functional $\mathcal{F}(p) = -\int p \log p, dx$. By the definition of the first-order variation, we have $\frac{\delta \mathcal{F}}{\delta p}(p) = -1 - \log p$, and therefore $\nabla \frac{\delta \mathcal{F}}{\delta p}(p) = -\nabla \log p$. This gradient term can be effectively estimated using standard score approximations; see [12].

# B Proof for Theorem 5.1

**Theorem 5.1** (Convergence guarantee of Flow Density Control with concave functionals). *Given Assumptions 5.1, fine-tuning a pre-trained model $\pi^{pre}$ via FDC (Algorithm 1) with $\eta_k = L \; \forall k \in [K]$, leads to a policy $\pi$ inducing a marginal distribution $p_1^\pi$ such that:*

$$\mathcal{G}(p_1^*) - \mathcal{G}(p_1^\pi) \leq \frac{L-l}{K} D_{KL}(p_1^* \,\|\, p_1^{pre}) \tag{12}$$

*where $p_1^* := p_1^{\pi^*}$ is the marginal distribution induced by the optimal policy $\pi^* \in \arg\max_\pi \mathcal{G}(p_1^\pi) := \mathcal{F}(p_1^\pi) - \alpha \mathcal{D}(p_1^\pi \,\|\, p_1^{pre})$.*

*Proof.* We prove this result using the framework of relative smoothness and relative strong convexity introduced in Section 5.

The analysis is based on the classical mirror descent framework under relative properties [33]. For notational simplicity, we let $\mu_k := p_T^{\pi_k}$, and fix an arbitrary reference density $\mu \in \mathbb{P}(\Omega_{\mathrm{pre}})$. To better align the notation of our theory with existing literature, we will proceed with the *convex* functional $\tilde{\mathcal{G}} := -\mathcal{G}$ below.

We begin by showing the following inequality:

$$\tilde{\mathcal{G}}(\mu_k) \leq \tilde{\mathcal{G}}(\mu_{k-1}) + \langle \delta\tilde{\mathcal{G}}(\mu_{k-1}), \mu_k - \mu_{k-1} \rangle + L D_{\mathcal{Q}}(\mu_k, \mu_{k-1}) \tag{33}$$

$$\leq \tilde{\mathcal{G}}(\mu_{k-1}) + \langle \delta\tilde{\mathcal{G}}(\mu_{k-1}), \mu - \mu_{k-1} \rangle + L D_{\mathcal{Q}}(\mu, \mu_{k-1}) - L D_{\mathcal{Q}}(\mu, \mu_k). \tag{34}$$

The first inequality follows from the $L$-smoothness of $\mathcal{G}$ relative to $\mathcal{Q}$ as defined in Definition 1. The second inequality uses the three-point inequality of the Bregman divergence [33, Lemma 3.1] with $\phi(\mu) = \frac{1}{L}\langle \delta\mathcal{G}(\mu_{k-1}), \mu - \mu_{k-1} \rangle$, $z = \mu_{k-1}$, and $z^+ = \mu_k$.

Next, using the $l$-strong concavity of $\mathcal{G}$ relative to $\mathcal{Q}$, again from Definition 1, we obtain:

$$\tilde{\mathcal{G}}(\mu_k) \leq \tilde{\mathcal{G}}(\mu) + (L-l) D_{\mathcal{Q}}(\mu, \mu_{k-1}) - L D_{\mathcal{Q}}(\mu, \mu_k). \tag{35}$$

By recursively applying the above inequality and using the monotonicity of $\mathcal{G}(\mu_k)$ along with the non-negativity of the Bregman divergence, we obtain [33]:

$$\sum_{k=1}^{K} \left(\frac{L}{L-l}\right)^k \left(\tilde{\mathcal{G}}(\mu_k) - \tilde{\mathcal{G}}(\mu)\right) \leq L D_{\mathcal{Q}}(\mu, \mu_0) - L\left(\frac{L}{L-l}\right)^K D_{\mathcal{Q}}(\mu, \mu_K) \leq L D_{\mathcal{Q}}(\mu, \mu_0). \tag{36}$$

Letting

$$\frac{1}{C_K} := \sum_{k=1}^{K} \left(\frac{L}{L-l}\right)^k, \tag{37}$$

and rearranging terms, we arrive at the convergence rate:

$$\tilde{\mathcal{G}}(\mu_K) - \tilde{\mathcal{G}}(\mu) \leq C_K L D_{\mathcal{Q}}(\mu, \mu_0) = \frac{l D_{\mathcal{Q}}(\mu, \mu_0)}{\left(1 + \frac{l}{L-l}\right)^K - 1}. \tag{38}$$

Finally, the convergence rate stated in the theorem follows by observing that $\left(1 + \frac{l}{L-l}\right)^K \geq 1 + \frac{Kl}{L-l}$.

$\square$

# C  Proof for Theorem 5.2

To establish our main convergence result, we introduce two additional technical assumptions that are satisfied in virtually all practical settings:

**Assumption C.1** (Support Compatibility). *We assume that the support of $p_T^{\pi_k}$ is contained in a fixed compact domain $\tilde{\Omega}$ for all $k$, and that for some $j$, we have $supp(p_j^{\pi_k}) = \tilde{\Omega}$.*

**Assumption C.2** (Precompactness). *The sequence $\{\delta\mathcal{H}(p_T^{\pi_k})\}_k$ is precompact in the topology induced by the $L_\infty$ norm.*

We are now ready to present the full proof. For the reader's convenience, we restate the theorem:

**Theorem 5.2** (Convergence guarantee of Flow Density Control for general functionals). *Given the Robbins-Monro step-size rule: $\sum_k \gamma_k = \infty, \sum_k \gamma_k^2 < \infty$, under Assumption 5.2 and technical assumptions (see Appendix C), the sequence of marginal densities $p_1^k$ induced by the iterates $\pi_k$ of Algorithm 1 converges weakly to a stationary point $\tilde{p}_1$ of $\mathcal{G}$ almost surely, formally: $p_1^k \rightharpoonup \tilde{p}_1$   a.s..*

*Proof.* We divide the proof into several key steps for ease of reading.

**Continuous-Time Mirror Flow.**  The main idea of our proof is to relate the discrete iterates $\{p_T^k\}_{k\in\mathbb{N}}$ produced by Algorithm 1 to a continuous-time mirror flow.

Define the initial dual variable as

$$h_0 = \delta\mathcal{H}(p_{\mathrm{pre}}) = -\log p_{\mathrm{pre}},$$

and consider the gradient flow

$$\begin{cases} \dot{h}_t = \delta\mathcal{G}(p_t), \\ p_t = \delta(-\mathcal{H})^\star(h_t), \end{cases} \tag{MF}$$

where $(-\mathcal{H})^\star(h) = \log\int_\Omega e^h$ is the Fenchel dual of the negative entropy functional [25, 22]. This defines the deterministic mirror flow associated with $\mathcal{G}$.

**Continuous-Time Interpolation of Iterates.**  To connect the discrete algorithm with (MF), we construct a continuous-time interpolation of the dual iterates $h^k = \delta\mathcal{H}(p_T^{\pi_k})$. Define the effective time

$$\tau^k = \sum_{r=0}^{k} \alpha_r,$$

and let the interpolated process $h(t)$ be

$$h(t) = h^k + \frac{t - \tau^k}{\tau + 1^k - \tau^k}(\tau + 1^k - h^k). \tag{Int}$$

Intuitively, our convergence result follows if two conditions hold:

**Informal Assumption 1** (Closeness to Continuous-Time Flow). *The interpolated process $h(t)$ asymptotically follows the dynamics of (MF) as $k \to \infty$.*

**Informal Assumption 2** (Convergence of the Flow). *The trajectories of (MF) converge to a stationary point of $\mathcal{G}$.*

To formalize this, we invoke the stochastic approximation framework of [5]. Let $\mathcal{Z}$ be the space of integrable functions on $\Omega$, and let $\Theta$ denote the flow of (MF). We define:

**Definition 2** (Asymptotic Pseudotrajectory (APT)). *We say $h(t)$ is an asymptotic pseudotrajectory (APT) of (MF) if for all $T > 0$,*

$$\lim_{t\to\infty} \sup_{0\le h\le T} \|h(t+h) - \Theta_h(h(t))\|_\infty = 0.$$

If $h(t)$ is a precompact APT, then [5] show:

**Theorem C.1** (APT Limit Set Theorem). *Let $h(t)$ be a precompact APT for the flow (MF). Then, almost surely, the limit set of $h(t)$ is contained in the set of internally chain-transitive (ICT) points of (MF).*

The proof of our result thus follows from two claims:

1. The iterates $\{h^k\}$ generate a precompact APT under Assumptions C.1 and 5.2.

2. The ICT set of (MF) consists only of stationary points of $\mathcal{G}$.

The remainder of the proof is devoted to verifying these two claims.

**Convergence to Stationary Points.** The second claim holds since the mirror flow admits $\mathcal{G}$ as a strict Lyapunov function, and thus Corollary 6.6 in [5] ensures convergence of the APT to the set of stationary points of $\mathcal{G}$, provided that the set of equilibria is countable.

For the first claim, Assumptions C.1 and C.2 ensure that the interpolated process is well-defined and precompact, while Assumption 5.2 allows us to apply standard stochastic approximation arguments [27]. We conclude the proof by applying Theorem C.1.

**Quantitative Approximation to the Mirror Flow.** For the first claim, we invoke the stochastic approximation techniques applied to the dual variables (see, e.g., [5, 27]) to obtain the following bound:

$$\sup_{0 \leq s \leq T} \|h(t+s) - \Theta_s(h(t))\| \leq C(T)\big[\Delta(t-1, T+1) + b(T) + \gamma(T)\big], \tag{39}$$

where $C(T)$ depends only on $T$, $\Delta(t-1, T+1)$ captures cumulative noise fluctuations, and $b(T), \gamma(T)$ are the bias and step-size terms over the interval. This explicitly bounds the deviation of the interpolated process from the deterministic mirror flow.

Under the noise and bias conditions in Assumption 5.2, standard stochastic approximation results [5, 27] imply

$$\lim_{T \to \infty} \Delta(t-1, T+1) = \lim_{T \to \infty} b(T) = 0.$$

Hence, $h(t)$ is an APT of the mirror flow.

**Conclusion.** Assuming precompactness of the dual iterates (stated as Assumption C.2), Theorem 5.7 in [5] implies that the limit set of $h(t)$ is internally chain transitive (ICT) for the mirror flow. Combining the quantitative approximation (39), the APT argument, and the limit set characterization, we conclude that the discrete iterates converge to stationary points of $\mathcal{G}$, completing the proof. $\square$

# D   Detailed Example of Algorithm Implementation

## D.1   Implementation of ENTROPYREGULARIZEDCONTROLSOLVER

To ensure completeness, below we provide pseudocode for one concrete realization of a ENTROPYREG-
ULARIZEDCONTROLSOLVER as in Eq. (8) using a first-order optimization routine. In particular, we de-
scribe exactly the version employed in Sec. 6, which builds on the Adjoint Matching framework [14],
casting linear fine-tuning as a stochastic optimal control problem and tackling it via regression.

Let $u^{pre}$ be the initial, pre-trained vector field, and $u^{finetuned}$ its fine-tuned counterpart. We also use
$\bar{\alpha}$ to refer to the accumulated noise schedule from [23] effectively following the flow models notation
introduced by Adjoint Mathing [14, Sec. 5.2]. The full procedure is in Algorithm 2.

---

**Algorithm 2** ENTROPYREGULARIZEDCONTROLSOLVER (Adjoint Matching [14]) based implementation

---

1: **Input:** $N$ : number of iterations, $u^{pre}$ : pre-trained flow vector field, $\eta$ regularization coefficient
   as in Eq. (8), $h$ : step size, $\nabla f$: reward function gradient, $m$ batch size
2: **Init:** $u^{finetuned} := u^{pre}$ with parameter $\theta$
3: **for** $n = 0, 1, 2, \ldots, N-1$ **do**
4:     Sample $m$ trajectories $\{X_t\}_{t=1}^T$ via memoryless noise schedule [14], e.g.,

$$\text{sample } \epsilon_t \sim \mathcal{N}(0, I), \ X_0 \sim \mathcal{N}(0, I), \text{ then:}$$

$$X_{t+h} = X_t + h\left(2v_\theta^{finetuned}(X_t, t) - \frac{\bar{\alpha}_t}{\alpha_t}X_t\right) + \sqrt{h}\sigma(t)\epsilon_t$$

Use reward gradient:

$$\tilde{a}_T = -\frac{1}{\eta}\nabla f(X_1)$$

For each trajectory, solve the lean adjoint ODE, see [14, Eq. 38-39], from $t = 1$ to $0$, e.g.,:

$$\tilde{a}_{t-h} = \tilde{a}_t + h\tilde{a}_t^\top \nabla_{X_t}\left(2u^{pre}(X_t, t) - \frac{\bar{\alpha}_t}{\alpha_t}X_t\right)$$

Where $X_t$ and $\tilde{a}_t$ are computed without gradients, i.e., $X_t = \texttt{stopgrad}(X_t), \tilde{a}_t = \texttt{stopgrad}(\tilde{a}_t)$. For each trajectory compute the Adjoint Matching objective [14, Eq. 37]:

$$\mathcal{L}_\theta = \sum_{t=0}^{1-h} \|\frac{2}{\sigma(t)}\left(u_\theta^{finetuned}(X-t, t) - u^{pre}(X_t, t)\right) + \sigma(t)\tilde{a}_t\|$$

Compute the gradient $\nabla_\theta \mathcal{L}(\theta)$ and update $\theta$.
5: **end for**
6: **output:** Fine-tuned noise predictor $u_\theta^{finetuned}$

---

## D.2   Discussion: computational complexity and cost of FDC

Flow Density Control (see Algorithm 1) is a sequential fine-tuning scheme, which performs $K$
iterations of a base fine-tuning oracle, as shown in Algorithm 1. Typically, as for the case of Adjoint
Matching [14], which is contextualized in Algorithm 2, the inner oracle also performs $N$ iterations to
solve the classic fine-tuning problem. As a consequence, at first glance, this lead to FDC having a
computational complexity scaling linearly in $K$ the one of classic fine-tuning. Nonetheless, this does
not seem to capture well the practical computational cost. In particular, we wish to point out the two
following observations:

- As discussed for the molecular design experiment in Sec. 6 and further in Appendix
  E, the FDC scheme might work well even with a very approximate oracle to solve the
  entropy-regularized control problem at each iteration.

- For many real-world problems a very small number of iterations $K$ might be sufficient to
  approximate the non-linear functional sufficiently well and hence obtain useful fine-tuned

models. This is shown in text-to-image bridge design experiment in Sec. 6 and in Appendix E. In this case, merely $K = 2$ iterations of FDC lead to promising results.

# E   Experimental Details

## E.1   Used computational resources

We run all experiments on a single Nvidia H100 GPU.

## E.2   Experiments in Illustrative Settings

**Shared experimental setup.**   For all illustrative experiments we utilize Adjoint Matching (AM) [14] for the entropy-regularized fine-tuning solver in Algorithm 1. Moreover, the stochastic gradient steps within the AM scheme are performed via an Adam optimizer.

**Risk-averse reward maximization for better worst-case validity or safety.**   In this experiment, we execute FDC for $K = 2$ iterations with a total of 1000 gradient steps within each iteration, AM solver (within the FDC scheme) with learning rate of $2e^{-2}$, $\alpha = 10^9$, and $\eta = 10$. Meanwhile, the AM baseline, is run for 1000 gradient steps with $\alpha = 0.2857$, and learning rate of $1e^{-5}$. The resulting CVaR is computed via the standard torch quantile method. The values of $\beta$ reported in the main paper effectively refers to the value of $1 - \beta$. In the following, we report mean and sample standard deviation of AM and FDC over 5 seeds.

|                    | $\text{CVaR}_\beta$ |
| ------------------ | ------------------- |
| Pre-trained        | $256.8 \pm 8.15$    |
| AM                 | $225.3 \pm 78.9$    |
| FDC (1 iteration)  | $221.1 \pm 73.2$    |
| FDC (2 iteration)  | $90.0 \pm 0.05$     |

Figure 5: Statistical analysis for $\text{CVaR}_\beta$.

**Novelty-seeking reward maximization for discovery.**   We run FDC for $K = 2$ iterations with a total of 1000 gradient steps within each iteration, AM solver (within the FDC scheme) with learning rate of $3e^{-6}$, $\alpha = 10^5$, and $\eta = 0.625$, and 8000 samples are used to estimate the first variation gradient as explained in Appendix A. Meanwhile, the AM baseline, is run for 1000 gradient steps with $\alpha = 0.666$, and learning rate of $1e^{-5}$. The resulting SQ is computed via the standard torch quantile method. In the following, we report mean and sample standard deviation of AM and FDC over 5 seeds.

|                    | $\text{SQ}_\beta$  |
| ------------------ | ------------------ |
| Pre-trained        | $59.6 \pm 7.5$     |
| AM                 | $56.7 \pm 2.7$     |
| FDC (1 iteration)  | $55.0 \pm 0.04$    |
| FDC (2 iteration)  | $452.5 \pm 250.0$  |

Figure 6: Statistical analysis for $\text{SQ}_\beta$ utility.

**Reward maximization regularized via optimal transport distance.**   Within this experiment, we present two runs of FDC, namely FDC-A and FDC-B, compared against AM. Both FDC-A and FDC-B have been run for $K = 6$ iterations of FDC, with $\alpha = 0.1$, AM oracle learning rate of $1e^{-6}$, $\eta = 6.666$. Both their discriminators to solve the dual OT problem as presented in Appendix A and mentioned within Sec. 4, have been learned via a simple MLP architecture with 800 gradient steps, by enforcing the 1-Lip. condition via the standard gradient penalty technique with regularization strength of $\lambda_{GP} = 10.0$ and learning rate of $1e^{-4}$. In particular, FDC-A is based on the distance defined, for two 2-dimensional points $x = (x_1, x_2)$ and $y = (y_1, y_2)$ by:

$$d_A(x, y) = \sqrt{(x_1 - y_1)^2 + (K(x_2 - y_2))^2}$$

Analogously, FDC-B leverages $d_B$ defined as:

$$d_A(x, y) = \sqrt{(K(x_1 - y_1))^2 + (x_2 - y_2)^2}$$

Where $K = 7$ in both cases. On the other hand, the AM baseline is run for $1000$ gradient steps with learning rate of $1e^{-3}$ and $\alpha = 1.538$.

**Conservative manifold exploration.** We ran FDC for $K = 50$ iterations and $2500$ gradient steps in total with $\eta = 10$ and $\alpha = 0.0, 0.01, 0.1, 0.5, 1.0$. We set the AM learning rate to $2e^{-4}$ and sample trajectories of length $400$ for computing the AM loss.

|       | $\mathbb{E}[r(x)]$ | $W_1^A$        | $\Delta\%$     |
| ----- | ------------------ | -------------- | -------------- |
| Pre   | $29.5 \pm 0.0$     | $0$            | $-$            |
| AM    | $35.08 \pm 0.04$   | $4.68 \pm 0.0$ | $100$          |
| FDC-A | $35.38 \pm 0.04$   | $1.92 \pm 0.03$| $288 \pm 15.0$ |

Figure 7: Statistical analysis for $W_1$ divergence.

### E.3 Further Ablations

**Runtime** The only input hyperparameter in Algorithm 1 is the number of iterations $K$. Towards evaluating its effect on algorithm execution, in the following, we consider an experimental setup analogous to "Risk-averse reward maximization for better worst-case validity or safety" experiment in Fig. 3 (top row), and evaluate the effect of different numbers of iterations ($K$) on run-time and solution quality. We report results in Fig. 8 showing the depending on hyper-parameter $K$ for $\eta = 20$. As one can expect, for small step-sizes $\frac{1}{\eta}$, the runtime and solution quality scale nearly linearly in $K$ given a fixed number of iterations $N$ of the entropy-regularized solver (see Apx. D for the definition of $N$). As one can expect by interpreting $\eta$ in Eq. 8 as a learning-rate parameter, by choosing

| K | Runtime (s) | CVaR estimate (via 1000 samples) |
| - | ----------- | -------------------------------- |
| 0 | 0.00        | 254.49                           |
| 1 | 44.71       | 241.37                           |
| 2 | 89.38       | 167.04                           |
| 3 | 133.31      | 288.72                           |
| 4 | 176.79      | 271.00                           |
| 5 | 220.37      | 84.89                            |
| 6 | 264.06      | 96.55                            |

Figure 8: Runtime vs. CVaR estimate as a function of $K$, $\eta = 20$.

smaller values of $\eta$ convergence can be achieved with less iterations $K$. In Fig. 9 we report the same evaluation with $\eta = 10$. The above tables hint at the fact that the FDC fine-tuning process can be

| K | Runtime (s) | CVaR estimate (via 1000 samples) |
| - | ----------- | -------------------------------- |
| 0 | 0.00        | 254.49                           |
| 1 | 44.23       | 249.09                           |
| 2 | 87.78       | 90.0                             |

Figure 9: Runtime vs. CVaR estimate as a function of $K$, $\eta = 10$.

interpreted experimentally as classic (convex or non-convex) optimization, although on the space of generative models, with learning rate (or step-size) controlled by $\eta$.

**Approximate Oracle** In the following, we investigate the use of an approximate entropy-regularized control solver oracle (i.e., performing approximate entropy-regularized fine-tuning at each iteration of FDC), showing that this can also lead to optimality via increasing the number of iterations $K$. In the following (see Fig. 10) we consider $N = 100$ instead of $N = 1000$ (as in previous experiments) and use $K = 5$ showing that FDC can retrieve the same final fine-tuned model as in Fig. 9 using only one tenth gradient steps (i.e. $N = 100$ instead of $N = 1000$) for the inner oracle.

| K | Runtime (s) | CVaR estimate (via 1000 samples) |
|---|---|---|
| 0 | 0.00 | 254.49 |
| 1 | 5.11 | 221.1 |
| 2 | 10.01 | 194.69 |
| 3 | 14.73 | 90.01 |
| 4 | 19.60 | 91.1 |
| 5 | 24.60 | 90.0 |

Figure 10: Runtime vs. CVaR estimate as a function of $K$, $\eta = 10$, $N = 100$.

## E.4   Real-World Experiments

**Molecular design for single-point energy minimization.**     In this experiment FDC is run for $K = 10$ iterations, with merely $2$ gradient steps at each iteration (i.e., the AM oracle is very approximate), AM learning rate of $1e^{-4}$, $\eta = 0.01$ and $\alpha = 0$. Meanwhile, the AM baseline is run for $240$ gradient steps with $\alpha = 0.0045$.

**Text-to-image bridge designs conservative exploration.**   For this experiment we ran FDC on a single Nvidia H100 GPU, with $K = 2$, $\eta = 200$, $\alpha = 0.001$ and a $100$ gradient steps in total. Similarly to previous work, we tuned the vector field resulting from applying classifier-free guidance with guidance scale $w = 8$ in SD-1.5.

