# OpenReview forum: "Flow Density Control: Generative Optimization Beyond Entropy-Regularized Fine-Tuning"
_NeurIPS.cc/2025/Conference — NeurIPS 2025 spotlight_

### Official Review · Reviewer_P6Lm · 2025-07-01

**Clarity:** 2
**Significance:** 3
**Originality:** 3
**Rating:** 4
**Confidence:** 4

**Summary:**

This paper introduces Flow Density Control (FDC), a method to fine-tune pre-trained flow /diffusion models beyond the standard expected-reward + KL formulation. This is achieved by leveraging recent machinery from Convex  and General Utilities RL.The authors prove convergence guarantees under ideal settings and give a proof outline of general settings.Experiments on 2D toy tasks, text-to-image generation, and molecular design shows that FDC can steer pre-trained models to solve tasks beyond the capabilities of current fine-tuning schemes.

**Questions:**

1.Gradient noise sensitivity: How robust is FDC when first-variation estimates (e.g., tail-quantile gradients) are noisy or biased?
2.Ablation on K and runtime: Can you report how performance and wall-clock time scale with the number of FDC iterations?
3.Add broader baselines:for instance, include a simple unregularized fine-tuning scheme and a policy-gradient method that directly targets tail-risk or high-quantile objectives.
4.Ablation on practical tasks: for instance,Inverse Imaging tasks and Editing & Conditional Generation tasks.

**Ethical Concerns:**

["NO or VERY MINOR ethics concerns only"]

**Final Justification:**

The authors’ rebuttal has allayed my initial concerns regarding theoretical rigor and experimental validation. They provided a much more detailed outline of the convergence proof, and the inclusion of runtime and sensitivity experiments strengthens the practical credibility of their approach, demonstrating that the method behaves as expected and that its claims are empirically grounded beyond the original limited discussion.

There are no major weaknesses in this paper, except for some minor points. So I suggest a weak acceptance recommendation for this paper.

**Limitations:**

Yes.

**Quality:**

3

**Strengths And Weaknesses:**

Strengths:
1. Clear derivation of FDC algorithm based on entropy-regularized control/RL, with convergence proofs (Theorems 5.1/5.2).
2. This framework supports a broad range of utility and divergence measures, addressing a gap in the generative model fine-tuning literature.
3. Experiments convincingly show risk-averse and novelty-seeking behavior; high-dim tasks show practical value.

Weaknesses:
1. Proof of Theorems 5.2 is insufficiently detailed and omits many of its required conditions. In particular, the interpolation closeness to the mirror flow is assumed but not quantified; the precompactness of dual iterates is stated (Assumption C.2) but not substantiated with uniform norm bounds; the APT noise conditions are not explicitly invoked; and the regularity of the mirror flow (MF)—required to ensure its gradient-flow structure and apply Sard’s theorem—is not rigorously justified. Given that convergence theory is a central contribution of the paper, these elements should be made explicit and backed by quantitative estimates.
2. Lack of enough ablation studies. Gradient of first variation estimate accuracy tolerance and hyperparameter sensitivity (e.g. iteraiton size K) are under discussed, raising questions about stability in real-world settings.

---

> ### Author Rebuttal · Authors · 2025-07-31
>
> We thank the Reviewer for recognizing our work as addressing a gap in the generative model fine-tuning literature, clear derivations of the algorithm, and experiments showing practical value. In the following, we address several points and questions mentioned within the review.
>
> **Proof of Theorem 5.2**
>
> We agree with the Reviewer that the current exposition does not present certain important details, which we will add for completeness and will revise the proof derivation accordingly. We provide here a sketch of the arguments and will elaborate further in the revised manuscript:
>
> - **Quantitative approximation to the mirror flow:**
>
>   Using stochastic approximation techniques applied to the dual variables, one can derive the following quantitative estimate (see, e.g., [1–3]):
>
>   $$
>   \sup_{0 \le s \le T} \|| h(t+s) - \Theta_s(h_t) \|| \le C(T) \bigl[\Delta(t-1,\,T+1) + b(T) + \gamma(T)\bigr],
>   $$
>
>   where $C(T)$ is a constant depending only on $T$, $\Delta(t-1, T+1)$ captures the cumulative noise fluctuations, and $b(T)$, \$\gamma(T)$ denote the bias and step‐size terms evaluated over the interval. This bounds the deviation of the interpolated process from the deterministic mirror flow.
>
> - **APT approximation:**
>
>   Under the noise and bias conditions stated in Assumption 5.2, it follows from standard results in stochastic approximation (cf. [1–3]) that both the noise fluctuation and bias vanish asymptotically:
>
>   $$
>    \lim_{T \to \infty} \Delta(t-1,\,T+1) = \lim_{T \to \infty} b(T) = 0.
>   $$
>
>   This establishes that the interpolated process is an **asymptotic pseudotrajectory (APT)** of the mirror flow.
>
> - **Limit set characterization:**
>
>   Given the precompactness of the dual iterates (which we state as an assumption but cannot presently prove from first principles), Theorem 5.7 in [1] implies that the limit set of the iterates is **internally chain transitive (ICT)** for the mirror flow.
>
> - **Avoiding Sard’s theorem:**
>
>   We note that invoking Sard’s theorem is not necessary in our argument. Since the continuous‐time mirror flow admits $\mathcal{G}$ as a strict Lyapunov function, Corollary 6.6 (which holds for an arbitrary metric space) in [1] applies and ensures convergence of the APT to the set of stationary points of $\mathcal{G}$, provided that the set of equilibria is *countable*. We thank the reviewer for the pointer and we will clarify this technical condition in the revision.
>
> We will integrate these details into the revised version, ensuring that all assumptions are clearly stated and each step in the convergence argument is substantiated with the appropriate estimates and references.
>
> **Ablation on $K$ and runtime**
> In the following, we consider the experimental setup analogous to the "Risk-averse reward maximization for better worst-case validity or safety" experiment in Fig. 3 (top row), and evaluate the effect of different numbers of iterations $K$ on run-time and solution quality, as well as evaluate the effect of a computationally cheaper approximate (entropy-regularized) fine-tuning oracle.
>
> - Dependency on hyper-parameter $K$ for $\eta = 20$. As one can expect, for small step-sizes $1$ / $\eta$, the runtime and solution quality improves nearly linearly in $K$ given a fixed number of iterations $N$ of the entropy-regularized solver (see Appendix D.1 for definition of $N$).
>
> |  **K** | **Runtime (s)** | **CVaR estimate (via 1000 samples)** |
> | :----: | :-------------: | :------------------------------: |
> |    0   |       0.00      |        254.48877624511718        |
> |    1   |      44.71      |        241.36963653564453        |
> |    2   |      89.38      |        167.03791809082030        |
> |    3   |      133.31     |        288.72082519531250        |
> |    4   |      176.79     |        271.00385437011720        |
> |    5   |      220.37     |         84.88514938354493        |
> |    6   |      264.06     |         96.55234909057617        |
>
> - As one can expect by interpreting $\eta$ in Eq. (8) as a learning-rate parameter, by choosing smaller values of $\eta$ convergence can be achieved with less iterations $K$. In the following, the same setup with $\eta =10$.
>
> | **K** | **Runtime (s)** | **CVaR estimate (1000 samples)** |
> |:----:|:---------------:|:---------------------------------:|
> | 0    | 0.00            | 254.48877624511718               |
> | 1    | 44.23           | 249.0867721557617                |
> | 2    | 87.78           | 90.0                             |
>
> The above two tables hint at the fact that the FDC fine-tuning process can be interpreted experimentally as classic (convex or non-convex) optimization, although on the space of generative models, with learning rate (or step-size) controlled by $\eta$.
>
> - Furthermore, using an approximate entropy-regularized control solver oracle (i.e., performing approximate entropy-regularized fine-tuning at each iteration of FDC) can also lead to optimality via increasing the number of iterations $K$. In the following we consider $N=100$ instead of $N=1000$ (as in previous experiments) and use $K=5$ showing that FDC can retrieve the same final fine-tuned model as in the previous table using only one tenth gradient steps (i.e. $N=100$ instead of $N=1000$) for the inner oracle.
>
> | **K** | **Runtime (s)** | **CVaR estimate (1000 samples)** |
> |:----:|:---------------:|:---------------------------------:|
> | 0    | 0.00            | 254.48877624511718               |
> | 1    | 5.11            | 221.07437133789062               |
> | 2    | 10.01           | 194.68543167114257               |
> | 3    | 14.73           | 90.01436312185169                |
> | 4    | 19.60           | 91.10320648061439                |
> | 5    | 24.60           | 90.0                             |
>
>
> **Gradient noise sensitivity**
> In the following, we consider an experimental setup analogous to "Risk-averse reward maximization for better worst-case validity or safety" experiment in Fig. 3 (top row), and evaluate the effect on solution quality of different numbers of samples for Monte Carlo estimation of the sample-based gradient of first variation (as presented in Sec. 4). Notice that $90.0$ corresponds to the optimal CVaR value in this experiment.
>
> | **Samples for MC estimate** | **CVaR estimate** |
> |:---------------------------:|:-----------------:|
> | 8000                        | 90.0              |
> | 4000                        | 90.0              |
> | 2000                        | 90.0              |
> | 1000                        | 124.0             |
> | 500                         | 90.0              |
> | 250                         | 212.2             |
> | 125                         | 90.0              |
> | 62                          | 90.0              |
> | 31                          | 154.9             |
>
> One can notice that in this experimental setup the method still converges to good solutions even for significantly lower amounts of samples than the one used for the experiments in the paper (i.e., $8000$). Further sample-complexity evaluations in real-world domains are significantly task and reward function dependent.
>
>
> **Further baselines and practical tasks**
> We believe that the baseline method we considered within our experimental evaluation, namely Adjoint Matching [4] is actually regarded by the community as the strongest, as well as widely adopted, available method to target the task at hand. More specifically, given a known reward function one whishes to maximize by fine-tuning a diffusion or flow model, there is significant literature regarding how to solve this problem (i.e. sample from the optimal reward-tilted distribution) via RL or Control theoretic techniques (e.g., see an overview in [5]). Early methods required to solve the problem approximately or with hidden value function bias (e.g. [6]) as discussed in [Section 4.2, 4], while recently Adjoint Matching [4] rendered possible to tackle the problem exactly and with the strongest empirical performances (see e.g. [4]) to our knowledge. Therefore, we regard it as a strong and expressive baseline, widely adopted, and already known to dominate alternative methods. If the Reviewer has in mind alternative specific baselines that are competitive with the one proposed we would be happy to hear and further discuss this point.
>
> **Inverse Imaging and Editing and Conditional Generation tasks**
> In this work, we tackle the problem of generative optimization. It is not particularly clear to us how the tasks mentioned by the Reviewer fit into the proposed generative optimization framework (in Sec. 3). Although interesting, we believe that exploring the connections of our framework with such problems and further evaluating the proposed method in other tasks goes beyond the scope of this work, which rather aims to introduce  novel algorithmic machinery beyond classic RL/Control schemes for generative optimization via fine-tuning.
>
> **References**
>
> [1] Benaim, Dynamics of Stochastic Approximation Algorithms, 1999
>
> [2] Mertikopoulos et al., A Unified Stochastic Approximation
> Framework for Learning in Games, 2024
>
> [3] Hsieh et al., The Limits of Min-Max Optimization Algorithms: Convergence to Spurious Non-Critical Sets, 2021
>
> [4] Domingo-Enrich et al., Adjoint Matching: Fine-tuning Flow and Diffusion Generative Models with Memoryless Stochastic Optimal Control. ICLR 2025.
>
> [5] Masatoshi Uehara et al., Understanding Reinforcement Learning-Based Fine-Tuning of Diffusion Models: A Tutorial and Review.
>
> [6] Masatoshi Uehara et al., Feedback efficient online fine-tuning of diffusion models. ICML 2024.

---

> > ### Comment · Reviewer_P6Lm · 2025-08-03
> >
> > 1. Rigor in the Revised Proof
> >
> > I appreciate that the authors will integrate these details into the revised paper. They have pledged to clearly state all assumptions and each step of the convergence proof with proper estimates and citations. This was a crucial missing piece in the submission, and their detailed rebuttal gives me confidence that this version will be far more rigorous.With these corrections, the theoretical foundation of FDC is significantly strengthened.
> >
> > 2. Runtime & Sensitivity Ablation Experiments
> >
> > Number of iterations (K): They show that increasing K steadily improves solution quality (e.g., lower CVaR) while scaling runtime linearly. The trade-off between inner solver fidelity and outer iterations is also analyzed, demonstrating flexibility in balancing compute and performance—an important practical insight.
> >
> > Gradient noise (Monte Carlo samples): Experiments varying the number of samples used in gradient estimation indicate that FDC remains robust under reduced precision. Performance degrades gracefully, confirming that extremely accurate gradients are not required in practice.
> >
> > Overall, these results strengthen the empirical case for FDC. While broader sensitivity analyses would still be beneficial, the added experiments increase confidence in the method’s stability and effectiveness across key hyperparameters.

---

> > > ### Author Response · Authors · 2025-08-06
> > >
> > > Dear Reviewer P6Lm,
> > >
> > > We are grateful to hear that our rebuttal clarified the concerns mentioned within the review both on the theoretical and experimental side. We will integrate such changes in the updated version of the work.
> > >
> > > Best Regards,
> > >
> > > Authors

---

> ### Comment · Area_Chair_FQdm · 2025-08-02
>
> Dear reviewer,
>
> Following the NeurIPS 2025 guidelines, I kindly encourage you to read and respond to author rebuttals as soon as possible. Please engage actively in the discussion with authors during the rebuttal process, update your review by filling in the "Final Justification" and acknowledge your involvement.
>
> Thank you,
> Your AC

---

### Official Review · Reviewer_Zmzc · 2025-07-01

**Clarity:** 3
**Significance:** 3
**Originality:** 3
**Rating:** 4
**Confidence:** 2

**Summary:**

This paper presents Flow Density Control (FDC), a new approach to fine-tuning pre-trained flow and diffusion generative models for a much broader class of objectives than previously possible. Existing work typically uses a linear expected-reward utility and KL-divergence regularization; FDC enables fine-tuning with general non-linear utilities (such as risk-averse, novelty-seeking, or diversity objectives) and more general divergences (e.g., optimal transport, Rényi).

**Questions:**

1. In line 244/245, why does the baseline method result in a worse loss than the basic model before fine-tuning?
2. In line 278, why can't we improve the rewards for the top 0.2% while maintaining the average reward?
3. Are there scenarios that require both risk-averse rewards and novelty-seeking rewards, and how do the methods perform in such contexts?

**Ethical Concerns:**

["NO or VERY MINOR ethics concerns only"]

**Limitations:**

Yes, but only a little

**Paper Formatting Concerns:**

NA.

**Quality:**

3

**Strengths And Weaknesses:**

## Strength
1. Theoretical Generalization: The paper rigorously extends finetuning of generative models to general utilities and divergences, going beyond standard entropy-regularized control. Section 3 formalizes this new generative optimization problem and classifies utility/divergence choices of practical interest.
2. The paper provides theoretical guarantees for the methodology.
3. The paper features a clear organizational structure and the figures are well-integrated and informative.


## Weakness
1. Although the showcased experiments are illustrative and varied, the empirical evaluation is still somewhat limited in scope. For the high-dimensional molecular and image tasks, more extensive quantitative metrics (e.g., additional standard baselines, statistical significance, generalization to other data sets or tasks) would strengthen the case that the generality of FDC yields practical upstream benefits in real-world applications.
2. The paper selects a very limited number of baseline comparison methods. However, as I am not familiar with the frontiers of this field, I need to refer to the opinions of other reviewers.
3. The paper fails to provide sufficient implementation details.
4. The discussion does not explicitly benchmark or analyze the computational overhead of sequentially running entropy-regularized control solvers versus standard finetuning. In large models or high-dimensional tasks, this could be a meaningful practical consideration.

---

> ### Author Rebuttal · Authors · 2025-07-31
>
> We thank the Reviewer for recognizing our work as rigorous, well organized, and with varied experiments, as well as appreciating its theoretical generalization of generative models fine-tuning. In the following, we address several points and questions mentioned within the review.
>
> **Further evaluation** We agree with the Reviewer regarding the relevance of statistical analysis. To this end, we report the estimated expectations and sample standard deviations for several experiments given by $5$ seeds. We consider experiments analogous to the first three types shown in the paper. These results confirm the insights within the submitted version of the paper and further validate the working mechanisms on the proposed method.
>
> - Risk-averse reward maximization for better worst-case validity or safety.
> | **Method**                                | **CVaR$_\beta$**          |
> |:-----------------------------------------|:-------------------------:|
> | Pre-trained                               | $256.8 \pm 8.15$         |
> | AM                                        | $225.3 \pm 78.9$         |
> | FDC (1 iteration)                         | $221.1 \pm 73.2$         |
> | FDC (2 iterations)                        | $90.0 \pm 0.05$          |
>
> - Novelty-seeking reward maximization for discovery.
> | **Method**                                | **SQ$_\beta$**           |
> |:-----------------------------------------|:------------------------:|
> | Pre-trained                               | $59.6 \pm 7.5$          |
> | AM                                        | $56.7 \pm 2.7$          |
> | FDC (1 iteration)                         | $55.0 \pm 0.04$         |
> | FDC (2 iterations)                        | $452.5 \pm 250.0$       |
>
> - Reward maximization regularized via optimal transport distance.
> | **Method**             | **$\mathbb{E}[r(x)]$**     | **$W_1^A$**        | **$\Delta\%$**        |
> |:-----------------------|:-------------------:|:------------------:|:---------------------:|
> | Pre                    | $29.5 \pm 0.0$      | $0$                | $-$                   |
> | AM                     | $35.08 \pm 0.04$    | $4.68 \pm 0.0$     | $100$                 |
> | FDC-A                  | $35.38 \pm 0.04$    | $1.92 \pm 0.03$    | $288 \pm 15.0$        |
>
> We thank the Reviewer for raising this point, and in an updated version of the work, we will add further statistical analysis of the proposed method as suggested by the Reviewer.
>
> **Baselines** We believe that the baseline method we considered within our experimental evaluation, namely Adjoint Matching (AM) [1] is actually regarded as the strongest, as well as widely adopted, available method to target the task at hand. More specifically, given a known reward function one whishes to maximize by fine-tuning a diffusion or flow model, there is significant literature regarding how to solve this problem via RL or Control theoretic techniques (e.g., see an overview in [2]). Early methods required to solve the problem approximately or with hidden value function bias (e.g., [3]) as discussed in [Section 4.2, 1], while recently AM [1] rendered possible to tackle the problem exactly and with the strongest empirical performances to our knowledge. Therefore, we regard it as a strong and expressive baseline, widely adopted, and already known to dominate alternative methods. If the Reviewer has in mind alternative specific baselines that are competitive with the one proposed we would be happy to hear and further discuss this point.
>
> **Implementation details** Alg. 1 is composed of two operations: (a) gradient estimation (line 4), and (b) fine-tuning via an entropy-regularized control solver (line 5). Gradient estimation is discussed in Sec. 4 (from line 166) for several common objectives, and in Appendix A.3 for the remaining ones. Meanwhile, a detailed implementation of the fine-tuning solver is reported in Appendix D.1 under "Detailed example of algorithm implementation". Nonetheless, we thank the Reviewer to mention this point and understand that it would be particularly helpful to find a specific complete implementation of FDC, which we will add in an updated version of the work.
>
> **Computational overhead** The computational overhead of FDC is discussed in Appendix D.2 under "Discussion: computational complexity and cost of FDC". Nonetheless, we agree with the Reviewer that its experimental analysis is relevant. To this end, in the following, we consider an experimental setup analogous to "Risk-averse reward maximization for better worst-case validity or safety" experiment in Fig. 3 (top row), and evaluate the effect of different numbers of iterations $K$ on run-time and solution quality, as well as evaluate the effect of a computationally cheaper yet approximate (entropy-regularized) fine-tuning oracle.
>
> - Dependency on hyper-parameter $K$ for $\eta = 20$. As one can expect, for small step-sizes $1$ / $\eta$, the runtime and solution quality improves nearly linearly in $K$ given a fixed number of iterations $N$ of the entropy-regularized solver (see Appendix D.1 for definition of $N$).
>
> |  **K** | **Runtime (s)** | **CVaR estimate (via 1000 samples)** |
> | :----: | :-------------: | :------------------------------: |
> |    0   |       0.00      |        254.48877624511718        |
> |    1   |      44.71      |        241.36963653564453        |
> |    2   |      89.38      |        167.03791809082030        |
> |    3   |      133.31     |        288.72082519531250        |
> |    4   |      176.79     |        271.00385437011720        |
> |    5   |      220.37     |         84.88514938354493        |
> |    6   |      264.06     |         96.55234909057617        |
>
> - As one can expect by interpreting $\eta$ in Eq. (8) as a learning-rate parameter, by choosing smaller values of $\eta$ convergence can be achieved with less iterations $K$. In the following, the same setup with $\eta =10$.
>
> | **K** | **Runtime (s)** | **CVaR estimate (1000 samples)** |
> |:----:|:---------------:|:---------------------------------:|
> | 0    | 0.00            | 254.48877624511718               |
> | 1    | 44.23           | 249.0867721557617                |
> | 2    | 87.78           | 90.0                             |
>
> - Furthermore, a cheaper entropy-regularized control solver can lead to optimality via increasing the number of iterations $K$. We consider $N=100$ instead of $N=1000$ (as in previous experiments) and use $K=5$. FDC can compute an optimal fine-tuned model using only one tenth solver gradient steps.
>
> | **K** | **Runtime (s)** | **CVaR estimate (1000 samples)** |
> |:----:|:---------------:|:---------------------------------:|
> | 0    | 0.00            | 254.48877624511718               |
> | 1    | 5.11            | 221.07437133789062               |
> | 2    | 10.01           | 194.68543167114257               |
> | 3    | 14.73           | 90.01436312185169                |
> | 4    | 19.60           | 91.10320648061439                |
> | 5    | 24.60           | 90.0                             |
>
> **Question 1** This is due to the fact that classic fine-tuning methods such as AM [1] cannot properly express the risk-averse objective (CVaR). In this case, fine-tuning the pre-trained model with AM to maximize expected rewards leads to a fine-tuned model more prone to generate worse designs in the worst-case. This can happen because expected reward maximization does not necessarily lead to optimization of the worst-case performances. On the contrary, FDC renders possible to directly control for the worst-case performance and trade-off average and worst-case generation quality of the fine-tuned model.
>
> **Question 2** Fine-tuning a generative model corresponds to moving a density to another area in space (or changing its distribution). As illustrated in Fig. 3 (middle row), it can happen that certain regions in space have higher average reward value while not presenting extraordinarily high 'discovery' regions (i.e., reward is high but with low variance). While other areas can have a lower average value, with small areas representing extraordinarily high reward sites (e.g., activity peaks in biological applications). A fine-tuned model that moves its density to the former region will have high average reward, while one fine-tuned to move to the latter area will have lower average reward, but potentially extraordinarily high reward for those few samples, or small amount of density, on the extraordinarily high reward sites. This is the standard setting where risk-seeking (or novelty-seeking) behaviour is necessary to discover the highest reward samples. Effectively, this can require sacrificing the average reward quality for the quality of the best performing samples.
>
> **Question 3** Scenarios of this type can occur and our framework can sharply capture this problem simply by considering a utility $\mathcal{F}$ defined as an arbitrarily weighted sum of the CVaR and SQ utilities in Table 1. This leads to a general GO problem (see Sec. 3, Fig. 2.b), alike SQ maximization, which can be tackled via Algorithm 1, and enjoying the theoretical guarantees presented in Sec. 5. In particular, the proposed method and theoretical guarantees are not tied to the sample of utilities and divergences presented in Table 1, but is general to any novel objective one wishes to maximize, either by composing known utilities or by novel mathematical expressions, as discussed in Sec. 3.
>
> **References**
>
> [1] Domingo-Enrich et al., Adjoint Matching: Fine-tuning Flow and Diffusion Generative Models with Memoryless Stochastic Optimal Control. ICLR 2025.
>
> [2] Masatoshi Uehara et al., Understanding Reinforcement Learning-Based Fine-Tuning of Diffusion Models: A Tutorial and Review.
>
> [3] Masatoshi Uehara et al., Feedback efficient online fine-tuning of diffusion models. ICML 2024.

---

> ### Comment · Area_Chair_FQdm · 2025-08-02
>
> Dear reviewer,
>
> Following the NeurIPS 2025 guidelines, I kindly encourage you to read and respond to author rebuttals as soon as possible. Please engage actively in the discussion with authors during the rebuttal process, update your review by filling in the "Final Justification" and acknowledge your involvement.
>
> Thank you,
> Your AC

---

> ### Comment · Area_Chair_FQdm · 2025-08-05
>
> Dear reviewer,
>
> Following the NeurIPS 2025 guidelines, I kindly encourage you to read and respond to author rebuttals **as soon as possible** as the reviewer-author discussion period is coming to the end. Please engage actively in the discussion with authors during the rebuttal process, update your review by filling in the "Final Justification" and acknowledge your involvement.
>
> Thank you,
> Your AC

---

> ### Author Response · Authors · 2025-08-06
>
> Dear Reviewer Zmzc,
>
> We believe our rebuttal addresses properly the concerns mentioned within the review by providing clear explanations as well as further experimental results regarding statistical analysis and parameter sensitivity. We kindly ask to let us know if any concern is still left unsolved.
>
> Best Regards,
>
> Authors

---

> > ### Comment · Reviewer_Zmzc · 2025-08-08
> >
> > Thank you for your detailed reply, which has basically addressed my concerns. I apologize for my delayed response. Taking into account the comments from other reviewers, I will maintain a positive score and increase the confidence level.

---

### Official Review · Reviewer_hbzy · 2025-07-02

**Clarity:** 4
**Significance:** 3
**Originality:** 3
**Rating:** 5
**Confidence:** 3

**Summary:**

The authors present a unified framework for optimizing general utilities by finetuning pre-trained flow models, subject to a constraint with respect to the pre-trained model (expressed by an arbitrary divergence). This framework generalizes prior work in fine-tuning flow/diffusion models, e.g., those that maximize expected reward subject to KL regularization. The authors present an algorithm, flow density control (FDC), to solve this general optimization problem, and prove convergence to a stationary point under mild assumptions. Finally, the authors conduct experiments in novelty-seeking molecular design and creativity-optimizing text-to-image generation.

**Questions:**

The main points that would cause me to increase my score are extensions to the experimental results: first and foremost, increased statistical rigor, and secondarily, more experiments showing the ability of FDC to "unlock" new capabilities in popular domains like text-to-image generation.

**Ethical Concerns:**

["NO or VERY MINOR ethics concerns only"]

**Final Justification:**

The authors have answered my questions and addressed my main concern about statistical rigor. I will maintain my recommendation to accept.

**Limitations:**

yes

**Quality:**

4

**Strengths And Weaknesses:**

Note: I do not have the theoretical background to check the derivations and proofs in detail.

**Strengths**

The paper is very well-written and very thorough. I understood exactly what problem the authors are trying to solve and why it's important. I do have some background in the theory of diffusion/flow models, specifically in fine-tuning them to maximize rewards (a special case of FDC), and I think FDC is a nice generalization of this idea. I appreciate the examples of various functionals presented in Table 1 --- even though not all of them are used in the experiments --- as well as the expanded explanations of their practical applications given in Appendix A. While I do not follow the theoretical analysis in detail, the primary result (Theorem 5.2) does seem strong --- the assumptions are fairly mild, and the authors are able to prove weak convergence to a stationary point of the utility functional.

Overall, the clarity and thoroughness of the paper provide a strong starting point for operationalizing FDC for many practical applications, and on this basis, I think this work would be a valuable contribution to the community.

**Weaknesses**

Though they are not the primary focus of the paper, the included experimental results are relatively weak. The authors do not include error bars or any measure of statistical certainty, citing lack of time and compute. I think this is a poor excuse, as sampling from a flow model is not so expensive compared to training. Tests for statistical certainty are important for scientific rigor. Also, the text-to-image experiment merely maximizes entropy subject to KL regularization, which is a capability already demonstrated in prior work.

---

> ### Author Rebuttal · Authors · 2025-07-31
>
> We thank the Reviewer for recognizing our work as very well-written and very thorough, the problem we are trying to solve as important, and our contribution as valuable to the community. In the following, we address several points and questions mentioned within the review.
>
> **Statistical analysis**
>
> We wish to point out that in order to perform a statistical analysis of the proposed fine-tuning method it would be needed to run such fine-tuning scheme multiple times rather than sampling multiple times. The latter is already done in the work for which we reported Monte Carlo estimates of expended performances (typically via approximately $5000$ samples). Nonetheless, we agree with the Reviewer regarding the relevance of statistical analysis. To this end, in the following we report the estimated expectations and sample standard deviations for several experiments given by $5$ seeds. In the following, we consider experimental setups analogous to the first three experiment types shown in the paper. These results confirm the insights already contained in the submitted version of the paper and further validate the working mechanisms of the proposed method.
>
> - Risk-averse reward maximization for better worst-case validity or safety.
> | **Method**                                | **CVaR$_\beta$**          |
> |:-----------------------------------------|:-------------------------:|
> | Pre-trained                               | $256.8 \pm 8.15$         |
> | AM                                        | $225.3 \pm 78.9$         |
> | FDC (1 iteration)                         | $221.1 \pm 73.2$         |
> | FDC (2 iterations)                        | $90.0 \pm 0.05$          |
>
> - Novelty-seeking reward maximization for discovery.
> | **Method**                                | **SQ$_\beta$**           |
> |:-----------------------------------------|:------------------------:|
> | Pre-trained                               | $59.6 \pm 7.5$          |
> | AM                                        | $56.7 \pm 2.7$          |
> | FDC (1 iteration)                         | $55.0 \pm 0.04$         |
> | FDC (2 iterations)                        | $452.5 \pm 250.0$       |
>
> - Reward maximization regularized via optimal transport distance.
> | **Method**             | **$\mathbb{E}[r(x)]$**     | **$W_1^A$**        | **$\Delta\%$**        |
> |:-----------------------|:-------------------:|:------------------:|:---------------------:|
> | Pre                    | $29.5 \pm 0.0$      | $0$                | $-$                   |
> | AM                     | $35.08 \pm 0.04$    | $4.68 \pm 0.0$     | $100$                 |
> | FDC-A                  | $35.38 \pm 0.04$    | $1.92 \pm 0.03$    | $288 \pm 15.0$        |
>
> We thank the Reviewer for raising this point, and in an updated version of the work, we will make sure to add further statistical analysis of the proposed method as suggested by the Reviewer.
>
> **Further ablation studies**
> Towards strengthening the experimental evaluation as suggested by the Reviewer, we conducted further experimental evaluations of the proposed method to analyse the effect on solution quality and runtime of hyperparameter choice and approximate (entropy-regularized) fine-tuning oracles. In the following, we consider an experimental setup analogous to the "Risk-averse reward maximization for better worst-case validity or safety" experiment in Fig. 3 (top row), and evaluate the effect of different numbers of iterations $K$ on run-time and solution quality.
> - Dependency on hyper-parameter $K$ for $\eta = 20$. As one can expect, for small step-sizes $1$ / $\eta$, the runtime and solution quality improves nearly linearly in $K$ given a fixed number of iterations $N$ of the entropy-regularized solver (see Appendix D.1 for definition of $N$).
>
> |  **K** | **Runtime (s)** | **CVaR estimate (via 1000 samples)** |
> | :----: | :-------------: | :------------------------------: |
> |    0   |       0.00      |        254.48877624511718        |
> |    1   |      44.71      |        241.36963653564453        |
> |    2   |      89.38      |        167.03791809082030        |
> |    3   |      133.31     |        288.72082519531250        |
> |    4   |      176.79     |        271.00385437011720        |
> |    5   |      220.37     |         84.88514938354493        |
> |    6   |      264.06     |         96.55234909057617        |
>
> - As one can expect by interpreting $\eta$ in Eq. (8) as a learning-rate parameter, by choosing smaller values of $\eta$ convergence can be achieved with less iterations $K$. In the following, the same setup with $\eta =10$.
>
> | **K** | **Runtime (s)** | **CVaR estimate (1000 samples)** |
> |:----:|:---------------:|:---------------------------------:|
> | 0    | 0.00            | 254.48877624511718               |
> | 1    | 44.23           | 249.0867721557617                |
> | 2    | 87.78           | 90.0                             |
>
> The above two tables hint at the fact that the FDC fine-tuning process can be interpreted experimentally as classic (convex or non-convex) optimization, although on the space of generative models, with learning rate (or step-size) controlled by $\eta$.
>
> - Furthermore, using an approximate entropy-regularized control solver oracle (i.e., performing approximate entropy-regularized fine-tuning at each iteration of FDC) can also lead to optimality via increasing the number of iterations $K$. In the following we consider $N=100$ instead of $N=1000$ (as in previous experiments) and use $K=5$ showing that FDC can retrieve the same final fine-tuned model as in the previous table using only one tenth gradient steps (i.e. $N=100$ instead of $N=1000$) for the inner oracle.
>
> | **K** | **Runtime (s)** | **CVaR estimate (1000 samples)** |
> |:----:|:---------------:|:---------------------------------:|
> | 0    | 0.00            | 254.48877624511718               |
> | 1    | 5.11            | 221.07437133789062               |
> | 2    | 10.01           | 194.68543167114257               |
> | 3    | 14.73           | 90.01436312185169                |
> | 4    | 19.60           | 91.10320648061439                |
> | 5    | 24.60           | 90.0                             |
>
>
>
> **Exploration in image experiments**
>
> To our knowledge, this is the first work that can perform KL-regularized entropy maximization, which we denote by Conservative Manifold Exploration, in a principled manner. Previous work (e.g., see [1]) could maximize entropy on the pre-trained model support only without any form of regularization. Crucially, such an un-regularized exploration objective risks to lead to an excessively explorative fine-tuned model that induces positive density in regions of invalid designs that do not correspond to any meaningful mode of the pre-trained model. On the other hand, the KL-regularized entropy objective that we introduced can render possible to balance the probability of sampling designs form diverse modes of the pre-trained model (even low-probability ones), while preventing sampling from regions of excessively low probability according to the prior model - thus 'Conservative', thanks to the divergence regularization term. This comparison is showcased in the "Conservative manifold exploration" experiment in Fig. 4 (top), where the method in [1] is retrieved for the sub-case of $\alpha = 0$. This also shows that the framework presented in our work is strictly more expressive than previous formulations for exploration via diffusion or flow models. If the Reviewer was referring to other works of which we are unaware, we are happy to further discuss once given more details.
>
> **'Unlocking new abilities' in text-to-image generation**
>
> The presented work proposes theoretically-grounded machinery to unlock promising abilities for this type of problems. Examples are CVaR maximization to prevent a text-to-image model to generate poor quality images in the worst case, according to a reward or preference model, SQ to generate particularly optimal images within a batch, and divergence regularized entropy-maximization for diverse text-to-image generation. Experimentally, we showcased the last application in the context of generation of bridge designs conditioned on a textual prompt showcasing increased diversity metrics (higher Vendi score) while maintaining high semantic quality score (CLIP). We regard this ability to generate more diverse images (due to the entropy term) while preserving their semantic quality (due to the divergence term) to be a novel and practically relevant ability for this class of models unlocked by the proposed scheme. We believe further applications of the proposed method on real-world problems, although both promising and interesting, go beyond the scope of this work.
>
> **References:**
>
> [1] De Santi et al., Provable maximum entropy manifold exploration via diffusion models. ICML 2025.

---

> > ### Comment · Reviewer_hbzy · 2025-08-04
> >
> > Thank you for the clarifications and additional results --- I especially appreciate the time taken to run 5 seeds. Regarding "unlocking new abilities", I suppose I just don't find the "diverse bridges" result that impressive, and perhaps a little underwhelming when contrasted with the promise of "unlocking new abilities". But again, it is not too big of a concern, and it's not the main focus of this paper.

---

> > > ### Author Response · Authors · 2025-08-06
> > >
> > > Dear Reviewer hbzy,
> > >
> > > We thank you for appreciating the main contributions of the paper as well as the new experiments within our rebuttal.
> > >
> > > Best Regards,
> > >
> > > Authors

---

> ### Comment · Area_Chair_FQdm · 2025-08-02
>
> Dear reviewer,
>
> Following the NeurIPS 2025 guidelines, I kindly encourage you to read and respond to author rebuttals as soon as possible. Please engage actively in the discussion with authors during the rebuttal process, update your review by filling in the "Final Justification" and acknowledge your involvement.
>
> Thank you,
> Your AC

---

### Official Review · Reviewer_LUT6 · 2025-07-02

**Clarity:** 2
**Significance:** 2
**Originality:** 3
**Rating:** 4
**Confidence:** 3

**Summary:**

This paper proposes Flow Density Control (FDC), a general framework for fine-tuning flow and diffusion generative models using a broad class of utilities and divergences. Extending existing works that mostly use KL-regularized rewards, FDC supports a broad class of objectives, including risk-averse, novelty-seeking, and various divergences.

**Questions:**

1. Given the motivation around molecular and biological design, expanding the biological experiments (e.g., multiple objectives, more evaluation metrics, realistic constraints) would significantly strengthen the practical contribution. Can the authors comment on this>

2. Please discuss how FDC compares with more recent non-KL regularized fine-tuning approaches such as those using Wasserstein, heavy-tailed distributions, or exploration-based objectives. I am not asking the authors to prove that this work can outperform novelty-seeking papers in novelty-seeking and can outperform risk-aware papers in controlling risks. I understand this work's major advantage is the generality of the schemework; at least, I would expect authors to discuss relevant works.

3. Can some of the non-linear objectives (e.g., CVaR, entropy maximization) be reformulated as expected reward maximization under transformed reward functions? (from my experience, I know CVaR can often be modeled via conditional expectation, so it is not principally different from other reward functions) If so, how does FDC differ in sample efficiency or convergence?

**Ethical Concerns:**

["NO or VERY MINOR ethics concerns only"]

**Final Justification:**

After reading the detailed response with abundant additional computational results, my confidence in the framework’s generality and rigor has improved a lot. I support the acceptance of this work.

**Quality:**

3

**Strengths And Weaknesses:**

Strength:

1. This work extends current methods by supporting general utilities (e.g., CVaR, entropy) and divergences (e.g., Rényi, OT).
2. Solid theoretical grounding with convergence analysis is also provided.
3. Experiments show benefits over prior methods in both synthetic and practical domains.

Weakness:

1. As a researcher with experience in AI for Science, I found the experiments, while extensive and strongly relevant, are a bit short in evaluation. First, molecular design on QM9 is quite limited as this dataset is rather small and is not representative enough nowadays. Have the authors consider experiments on large datasets such as zinc250k？ Second, the discrete generative modeling context is especially relevant in biological design, for example, DNA sequences and proteins. But any sort of biology experiment is not included in experiments?

2. The writing of the experiment section is too concise. Many necessary details are not provided. I found that section hard to parse. Besides, it seems ablation studies and sensitivity analysis for hyperparameters are also missing.

---

> ### Author Rebuttal · Authors · 2025-07-31
>
> We thank the Reviewer for recognizing our work as with solid theoretical grounding as well as with extensive and strongly relevant experiments. In the following, we address several points and questions mentioned within the review.
>
> **Biology experiments**
>
> In this work, we propose a novel probability-space optimization framework that enables fine‑tuning of flow and diffusion models for a broader class of utilities beyond the reach of classic control or RL methods. While this foundational contribution is significantly motivated by scientific discovery applications, showing impact in such applications goes beyond the scope of this work. The presented sample application on QM9, a computational chemistry setup, although not capturing the full complexity of modern chemical and biological tasks, shows that the proposed theoretically-grounded method is indeed (very) promising for biological discovery tasks, where we hope to see it tested as a natural next step.
>
> **Discrete generative modeling**
>
> While the focus of this work is specific to continuous-state flow and diffusion models, which cover many practically relevant biological discovery applications (e.g. protein docking for drug discovery [5]), we believe the algorithm proposed can be easily extended to control discrete diffusion or flow models as well. This is due to the recent introduction of entropy-regularized fine-tuning oracles for such discrete models (see, e.g. Eq. (5) in [1] for discrete diffusion), which can be employed to implement our method for such discrete models.
>
> **Experiments section too concise**
>
> We agree with the Reviewer: unfortunately this was due to page limits. We plan to use the extra page of the camera-ready version to extend the experimental section with further experimental details as well as further ablation studies as reported in the following point.
>
> **Further ablation and sensitivity analysis**
>
> We have shown promising performances of the proposed method for six different utilities and/or tasks of practical relevance. Since the algorithm proposed is particularly principled, its dependency on hyperparameters can be inferred from its (convex) optimization interpretation and is partially discussed in Appendix D.2 under "Discussion: computational complexity and cost of FDC". Nonetheless, we agree with the Reviewer that the experimental section can be further strengthened via ablation studies and sensitivity analysis wrt the algorithm hyperparameters. We wish to point out that our method can leverage as oracle any entropy-regularized fine-tuning method, such as Adjoint Matching [2], and the practical performance of FDC significantly depends on the one of such oracles. Luckily, there exist several oracles with established evaluations in complex settings (see, e.g., [2] for images or [6] for molecules and proteins). Effectively, the only input hyperparameter shown in Algorithm 1 is the number of iterations $K$. Towards evaluating its effect on algorithm execution, in the following, we consider an experimental setup analogous to "Risk-averse reward maximization for better worst-case validity or safety" experiment in Fig. 3 (top row), and evaluate the effect of different numbers of iterations \(K\) on run-time and solution quality. Notice that in the following tasks the optimal CVaR value is $90.0$.
> - Dependency on hyper-parameter $K$ for $\eta = 20$. As one can expect, for small step-sizes $1$ / $\eta$, the runtime and solution quality improves nearly linearly in $K$ given a fixed number of iterations $N$ of the entropy-regularized solver (see Appendix D.1 for definition of $N$).
>
> |  **K** | **Runtime (s)** | **CVaR estimate (via 1000 samples)** |
> | :----: | :-------------: | :------------------------------: |
> |    0   |       0.00      |        254.48877624511718        |
> |    1   |      44.71      |        241.36963653564453        |
> |    2   |      89.38      |        167.03791809082030        |
> |    3   |      133.31     |        288.72082519531250        |
> |    4   |      176.79     |        271.00385437011720        |
> |    5   |      220.37     |         84.88514938354493        |
> |    6   |      264.06     |         96.55234909057617        |
>
> - As one can expect by interpreting $\eta$ in Eq. (8) as a learning-rate parameter, by choosing smaller values of $\eta$ convergence can be achieved with less iterations $K$. In the following, the same setup with $\eta =10$.
>
> | **K** | **Runtime (s)** | **CVaR estimate (1000 samples)** |
> |:----:|:---------------:|:---------------------------------:|
> | 0    | 0.00            | 254.48877624511718               |
> | 1    | 44.23           | 249.0867721557617                |
> | 2    | 87.78           | 90.0                             |
>
> The above two tables hint at the fact that the FDC fine-tuning process can be interpreted experimentally as classic (convex or non-convex) optimization, although on the space of generative models, with learning rate (or step-size) controlled by $\eta$.
>
> - Furthermore, using an approximate entropy-regularized control solver oracle (i.e., performing approximate entropy-regularized fine-tuning at each iteration of FDC) can also lead to optimality via increasing the number of iterations $K$. In the following we consider $N=100$ instead of $N=1000$ (as in previous experiments) and use $K=5$ showing that FDC can retrieve the same final fine-tuned model as in the previous table using only one tenth gradient steps (i.e. $N=100$ instead of $N=1000$) for the inner oracle.
>
> | **K** | **Runtime (s)** | **CVaR estimate (1000 samples)** |
> |:----:|:---------------:|:---------------------------------:|
> | 0    | 0.00            | 254.48877624511718               |
> | 1    | 5.11            | 221.07437133789062               |
> | 2    | 10.01           | 194.68543167114257               |
> | 3    | 14.73           | 90.01436312185169                |
> | 4    | 19.60           | 91.10320648061439                |
> | 5    | 24.60           | 90.0                             |
>
> We thank the Reviewer for the question and will certainly discuss these dependences on hyperparameters as well as oracle quality in an updated version of the work.
>
>
>
> **Comparison with related works for fine-tuning of general utilities**
>
> To the best of our knowledge, the only work that directly tackles a specific instance of convex or general utilities as depicted within our work is [3], which tackles the sub-case of entropy maximization and we compare in our work. In particular, in the "Conservative manifold exploration." experiment, we explain how our framework can extend entropy maximization introduced in [3] to a "conservative" approach where a divergence measure from the prior model can be enforced, which is not possible in prior work. To our knowledge, no other work that aims to control flow models via continuous-time RL, or stochastic control, shows how to tackle similar general objectives. Other approaches seem to tackle specific sub-cases via approximating such complex non-linear objectives and then use classic RL machinery (e.g.,  [4]). In comparison to such schemes, our work renders possible to directly optimize non-linear objectives without the need of developing explicit and objective-specific approximations, which do not necessarily exist and that can be arbitrarily loose. We thank again the Reviewer for the question and we will further clarify these connections in the revised version of the manuscript.
>
> **Objectives reformulation**
>
> We are not aware of such cases. The CVaR is typically defined as a conditional expectation (as in Table 1 in our work) rendering it linear given the quantile. But the latter quantity depends non-linearly on the measure rendering the CVaR non-linear by definition. For this reason, it is typically treated as a convex (or non-convex, depending on its parametrization) functional in related works (see, e.g., Table 1 in [7] or [8]). If the Reviewer can provide further details about this point, we would be happy to further discuss.
>
> **References**
>
> [1] Wang et al., Fine-tuning discrete diffusion models via reward optimization with applications to dna and protein design. ICLR 2025.
>
> [2] Domingo-Enrich et al., Adjoint Matching: Fine-tuning Flow and Diffusion Generative Models with Memoryless Stochastic Optimal Control. ICLR 2025.
>
> [3] De Santi et al., Provable maximum entropy manifold exploration via diffusion models. ICML 2025.
>
> [4] Fan et al., Online Reward-Weighted Fine-Tuning of Flow Matching with Wasserstein Regularization. ICLR 2025.
>
> [5] Corso et al., Diffdock: Diffusion steps, twists, and turns for molecular docking. ICLR 2023.
>
> [6] Uehara et al., Feedback efficient online fine-tuning of diffusion models. ICML 2024.
>
> [7] Mutti et al., Convex Reinforcement Learning in Finite Trials. JMLR 2023.
>
> [8] Mutti et al., Challenging Common Assumptions in Convex Reinforcement Learning. NeurIPS 2022.

---

> ### Comment · Area_Chair_FQdm · 2025-08-02
>
> Dear reviewer,
>
> Following the NeurIPS 2025 guidelines, I kindly encourage you to read and respond to author rebuttals as soon as possible. Please engage actively in the discussion with authors during the rebuttal process, update your review by filling in the "Final Justification" and acknowledge your involvement.
>
> Thank you,
> Your AC

---

> ### Comment · Area_Chair_FQdm · 2025-08-05
>
> Dear reviewer,
>
> Following the NeurIPS 2025 guidelines, I kindly encourage you to read and respond to author rebuttals **as soon as possible** as the reviewer-author discussion period is coming to the end. Please engage actively in the discussion with authors during the rebuttal process, update your review by filling in the "Final Justification" and acknowledge your involvement.
>
> Thank you,
> Your AC

---

> ### Comment · Reviewer_LUT6 · 2025-08-05
>
> I thank the authors for drafting the detailed response and for providing additional computational results. The newly added clarifications on scope, hyperparameter sensitivity, and related work were helpful. The rebuttal improves my confidence in the framework’s generality and rigor. I would encourage the authors to complement the next version with these discussions. If possible, I still argue that broader biological evaluations and clearer experimental clarifications are beneficial for the next version of this work.

---

> > ### Author Response · Authors · 2025-08-06
> >
> > Dear Reviewer LUT6,
> >
> > We are grateful to hear that our rebuttal has improved your confidence in the framework’s generality and rigor, and we thank you for your valuable suggestions.
> >
> > Best Regards,
> >
> > Authors

---

### Decision · Program_Chairs · 2025-09-17

**Decision:**

Accept (spotlight)

**Comment:**

This paper considers the problem to fine-tune pre-trained flow models. The authors extended the mostly studied KL-regularized reward fine-tuning setting to a more general reward and regularization with theoretical analysis. The proposed approach was applied on a variety of settings and problems including toy problem, text-to-image generation and molecular design.

All the reviewers vote to accept this paper. Initially, the reviewers mostly questioned about the theoretical rigor and experiment section and the authors included more analysis and experimental results that convinced the reviewers.

In particular, I find the generalization considered in this paper nice and the great effort showed in designing experiments/applications. I vote to accept this paper as a spotlight presentation because it is worth for more audience to consider this framework from the list of applications and reward/regularization design. I would also encourage the authors to consider more application settings to expand the broader impact of the work.